# SpikingLLM: Spiking Large Language Models with Causal Spiking Self-Attention and Spike-Form Knowledge Distillation

## Abstract

Spiking Neural Networks (SNNs) offer promising energy-efficient alternatives to large language models (LLMs) due to their event-driven nature and ultra-low power consumption. However, to retain representation capacity, most existing spiking LLM approaches rely on integer activations or softmax, which involve intensive floating-point operations and undermine inference efficiency. Moreover, the intrinsic spatial-temporal optimization of spiking networks further increase the direct training cost and difficulty. To address these challenges, we propose **SpikingLLM**, the first fully binary spike-driven spiking LLM framework developed from random initialization, without reliance on floating-point matrix multiplications or softmax. At the core of SpikingLLM is the **Causal Spiking Self-Attention (CSSA)** mechanism, which replaces conventional softmax with binary spike-based operations and thereby enables autoregressive language modeling in the spiking domain, ensuring low-cost inference. To support cost-efficient training under constrained computational budgets, we further introduce **Spike-Form Knowledge Distillation (SKD)**, a multi-level distillation strategy that aligns ANN teacher and SNN student across embeddings, attention maps, intermediate features, and output logits. SKD framework allows SpikingLLM to achieve competitive performance with ANN counterparts using substantially fewer training tokens (e.g., 1.0B tokens for a 0.125B model and 10.0B tokens for a 1.3B model), resulting in effective training. As a result, SpikingLLM achieves ANN-level performance at only **4.16%–5.87%** of the computational cost on natural language generation tasks. Our results highlight the feasibility and effectiveness of fully binary spike-driven LLMs and establish the distillation as a promising pathway for energy-efficient, brain-inspired spiking NLP.

## 1 Introduction

Large Language Models (LLMs) have demonstrated remarkable capabilities in natural language processing, powering a wide range of applications from conversational agents to code generation (Brown et al., 2020; Achiam et al., 2023). However, these models typically require extensive computational resources and energy consumption during both training and inference. For example, GPT-3 was trained with 175 billion parameters using hundreds of petaflop/s-days of compute (Brown et al., 2020). In addition, inference also incurs substantial energy costs, as serving a single query can involve billions of operations and significant GPU utilization (Strubell et al., 2020; Schwartz et al., 2020), raising concerns about their scalability and environmental impact (Strubell et al., 2020).

Compared with ANN-based LLMs, the human brain achieves superior intelligence with drastically lower energy consumption, operating on just 20 watts to power approximately 86 billion neurons (Izhikevich, 2003; Gerstner et al., 2014). Inspired by the brain's energy-efficient signaling, Spiking Neural Networks (SNNs) (Maass, 1997; Gerstner et al., 2014) communicate through binary spike events, enabling event-driven and low-power computation (Yin et al., 2021; Schuman et al., 2022), making SNNs a promising alternative to traditional ANNs.

While recent efforts have shown promising results of SNNs in computer vision tasks (Zhou et al., 2024; Li et al., 2024; Luo et al., 2024), extending SNNs to natural language processing (NLP),

Figure 1: Overview of **Causal Spiking Self-Attention (CSSA)**. Left: Comparison between Vanilla Causal Self-Attention (CSA) (bottom) and CSSA (top). CSA uses softmax and additive masks, while CSSA employs spike-based activation and binary causal masking. Right: Detailed CSSA pipeline, showing spike-form Q, K, V computation, masked integer attention, spiking activation, and spike-based output, enabling fully discrete and energy-efficient attention modeling.

especially LLMs, remains largely underexplored. A central challenge is the design of spiking attention mechanisms. In contrast to vision models, where representations are often bidirectional and spatially local, autoregressive LLMs require **causal attention** to ensure that each token prediction depends only on its preceding context. However, conventional causal attention relies on floating-point matrix multiplications and the softmax operation, both of which are computationally intensive and fundamentally incompatible with spike-based processing. Existing attempts either retain these components (Zhu et al., 2023) or introduce multi-threshold neurons and integer activations (Xing et al., 2024a), which still incur substantial floating-point overhead. Designing a spike-driven causal attention mechanism is therefore critical: it must eliminate softmax while preserving the autoregressive representational capacity of binary spike trains. This challenge directly motivates our **Causal Spiking Self-Attention (CSSA)**, which enables efficient spike-based sequence modeling for spiking LLMs.

Moreover, training spiking LLMs introduces additional difficulties beyond those in vision tasks. The inherent temporal dynamics of SNNs already leads to complex computational graphs and high computational cost during backpropagation. Scaling up the architecture further exacerbates this, making full end-to-end training inefficient or even infeasible. Consequently, prior works mostly resort to ANN-to-SNN conversions (Xing et al., 2024a; Schmidgall et al., 2024). However, such methods typically require large time steps to approximate ANN activations, resulting in high inference cost. Integer-based conversions further scale the operations by $T \times N$, which compromises the potential energy benefits of event-driven spiking computation.

To address these challenges, we propose SpikingLLM, a spike-based large language model built on two key components: a spike-driven attention mechanism (CSSA), schematically depicted in Figure 1, and a multi-level knowledge distillation scheme (SKD), presented in Figure 3. Overall, our contributions can be summarized as follows:

- We propose **SpikingLLM**, the spike-based large language model equipped with a fully spike-driven attention mechanism. Our **CSSA** replaces the vanilla causal self-attention, which relies on floating-point operations and softmax, with a spike-based computation, enabling efficient autoregressive sequence modeling with binary spikes. The overall design follows the OPT-family architecture (Zhang et al., 2022), adapted to the spiking domain.

- We introduce **SKD**, a novel training framework that enables SpikingLLM to be directly trained from random initialization. SKD distills multi-level knowledge covering embeddings, attention maps, intermediate features, and output logits from the teacher model,

thereby accelerating convergence, improving training stability, and reducing the amount of training data required for large-scale spiking LLMs.

- With only 10B training tokens, significantly fewer than the 180B tokens used to train OPT-1.3B, our SKD framework enables SpikingLLM-1.3B to achieve 42.19% zero-shot accuracy on common reasoning benchmarks using 4 time steps, approaching the 49.73% of OPT-1.3B, while consuming just 10.6% of the energy per inference. Remarkably, even at 2 time steps, the model maintains 41.33% accuracy with only 5.88% of the energy cost.

## 2 RELATED WORK

### 2.1 SNNS IN DOWNSTREAM TASKS

Recent works show SNNs achieving competitive performance in vision tasks with lower computational consumption. In image classification, advances in surrogate gradients, attention, and adaptive thresholds have boosted accuracy and efficiency on CIFAR-10/100 and ImageNet (Rathi et al., 2020; Zhou et al., 2022; 2023; 2024; Li et al., 2024). In object detection, models like SFOD and attention-based SNNs reduce energy cost while closing the gap with ANNs (Su et al., 2023; Bodden et al., 2024; Li et al., 2025). For event-based vision, architectures such as 3D-SNN, and MG-SNN effectively handle gesture, motion, and optical flow tasks (Orchard et al., 2015; Lee et al., 2020; Gehrig et al., 2021).

In contrast, the application of SNNs in natural language processing (NLP) is still largely underexplored, with only a few attempts adapting language models to spike-based computation. For example, Lv et al. (2023) employs a two-stage distillation strategy to align a pre-trained BERT with an SNN, but retains many floating-point operations and is limited in scale (up to 109M parameters). Xing et al. (2024b) proposes a spike-driven language model with bi-directional encoding, yet relies on floating-point spikes and retains dense floating-point operations, undermining the event-driven efficiency. Zhu et al. (2023) replaces attention with a linear-complexity Spiking RWKV module, but still depends on dense floating-point computation and remains modest in size (216M parameters). Xing et al. (2024a) pushes scaling further by introducing the GIF neuron and OBSpiking framework, enabling model sizes from 7B to 70B. However, this strategy substitutes binary spike trains with quantized integer signals and retains the softmax operation, thereby losing the advantages of event-driven computation and fine-grained temporal dynamics. Additionally, despite targeting autoregressive large language modeling, the attention mechanism does not incorporate causal masking adapted to SNN timing constraints.

### 2.2 KNOWLEDGE DISTILLATION

Knowledge distillation is a widely adopted approach for compressing large-scale language models into smaller, more efficient ones, as demonstrated by models like DistilBERT and TinyBERT (Sanh et al., 2019; Jiao et al., 2019). In the context of SNNs, early distillation efforts have primarily targeted small-scale vision tasks, using spike-based student networks guided by soft targets from ANN teachers (Xu et al., 2023; Qiu et al., 2024; Xu et al., 2024).

In contrast, spike-based distillation for language modeling remains underexplored. Existing methods often overlook the temporal dynamics of SNNs or lack alignment in the spike domain. For example, SpikeBERT (Lv et al., 2023) maps spike activations into continuous representations via an additional MLP for teacher-student alignment. However, this introduces extra trainable parameters and computational overhead, while bypassing the native spike representation, thus limiting the preservation of spike-driven semantics. To address this, we propose the Spike-Form Knowledge Distillation framework tailored for SpikingLLMs, featuring spike-attention and spike-feature alignment modules that enables multi-level knowledge transfer while preserving the discrete and temporal nature of spiking computation.

## 3 METHODS

We propose SpikingLLM, a spike-based large language model that integrates a spike-native architectural design with an efficient training paradigm tailored for large-scale SNNs. Specifically, we

Figure 2: Left depicts the SpikingLLM framework, detailing the operations of the Causal Spiking Self-Attention (CSSA) module and the Spiking Feed-Forward Network (SFFN). Right compares the computational process of vanilla Causal Self-Attention (CSA), Spiking Self-Attention (SSA), and CSSA, where red spikes represent binary values of 1 and all other values are 0.

design a fully spike-driven attention mechanism, Causal Spiking Self-Attention (CSSA), which replaces conventional softmax-based attention with spike-compatible computation, supporting autoregressive sequence modeling using binary spikes. Building upon this architecture, we further develop Spike-Form Knowledge Distillation (SKD), a multi-level distillation framework that enables stable and scalable training from random initialization by transferring rich supervision signals from a frozen ANN teacher to the SNN student. The overall model architecture is shown in Figure 2, and the training strategy is illustrated in Figure 3.

## 3.1 PROBLEM STATEMENT

We consider the task of autoregressive generation using a decoder-only Large Language Model (LLM). Formally, given a sequence of tokens $x_1, x_2, \ldots, x_n$, the model is trained to predict the next token $x_{n+1}$ conditioned on the previous $n$ tokens. This can be expressed as maximizing the likelihood:

$$P(x_{n+1} \mid x_1, x_2, \ldots, x_n). \tag{1}$$

During the pre-training stage, the ground-truth label for each autoregressive generation step $tau$ is the token $x_{tau+1}$, and the model is optimized using the standard cross-entropy loss. The goal is to learn a function that maps token sequences to probability distributions over the vocabulary, employing causal (unidirectional) attention under temporal constraints.

## 3.2 SPIKINGLLM ARCHITECTURE

To enable efficient sequence modeling with SNNs, we propose **SpikingLLM**, which integrates binary spiking neurons with causal attention for softmax-free, energy-efficient computation. Unlike prior works (Zhu et al., 2023; Xing et al., 2024a; Schmidgall et al., 2024), SpikingLLM is fully spike-driven and employs a Hadamard-masked dot product followed by spiking neuron to implement causal attention without softmax. The architecture consists of three main components: (1) Spiking Neuron Modules, (2) Causal Spiking Self-Attention (CSSA), and (3) a Spike Feed-Forward Network (SFFN). The overall design is built upon the OPT-family architecture (Zhang et al., 2022), chosen for its open-source nature, simplicity, and proven effectiveness. The model is further adapted to operate entirely with spiking computations.

### 3.2.1 SPIKING NEURON MODULES

To explore more expressive yet efficient spiking neurons for language modeling, we design two variants of SpikingLLM. **SpikingLLM-v1** employs the standard Leaky Integrate-and-Fire (LIF) neuron (Wu et al., 2018), implemented via SpikingJelly (Fang et al., 2023), while **SpikingLLM-v2** adopts a ternary spiking neuron inspired by (Xing et al., 2024b), which extends binary spikes $\{0, 1\}$ to ternary values $\{-\alpha, 0, +\alpha\}$ depending on the membrane potential intensity.

The LIF neuron emits a spike $S_t \in \{0, 1\}$ when the membrane potential $U_t$ exceeds a threshold $U_{thr}$, and resets afterward:

$$S_t = \begin{cases} 1, & \text{if } U_t \geq U_{thr}, \\ 0, & \text{otherwise}, \end{cases} \quad U_t = I_t + \lambda U_{t-1} - S_{t-1} U_{thr}, \tag{2}$$

where $I_t = W X_t$ is the input current, and $\lambda$ controls temporal decay.

In contrast, the ternary neuron in SpikingLLM-v2 outputs discrete values scaled by a layer-specific amplitude $\alpha(t)$:

$$s_{\pm}(t) = \begin{cases} -\alpha(t), & \text{if } m(t) < -\alpha(t), \\ 0, & \text{if } |m(t)| \leq \alpha(t), \\ +\alpha(t), & \text{if } m(t) > +\alpha(t), \end{cases} \tag{3}$$

with membrane potential updated as:

$$v_l(t) = m_l(t)(\alpha(t) - s_l(t)) + v_{\text{reset}} s_l(t). \tag{4}$$

While SpikingLLM-v2 captures richer signal representations, it introduces additional computation and deviates from the strict sparsity and event-driven efficiency of binary SNNs.

### 3.2.2 CAUSAL SPIKING SELF-ATTENTION (CSSA)

To enable attention mechanisms in spike-based neural networks while preserving computational efficiency, we propose the **Causal Spiking Self-Attention (CSSA)** module, presented in Figure 1. CSSA reformulates the classical self-attention mechanism using spike-based representations, constrained by causality and spiking dynamics.

Specifically, input spike sequences are first projected into continuous-valued queries, keys, and values, which are then discretized via LIF or ternary spiking neurons. Spike-based dot products between queries and keys yield integer-valued attention scores, followed by a causal mask to ensure autoregressive flow. The masked scores are passed through a spiking activation to produce sparse attention weights, which are used to compute the weighted sum over value spikes. A final linear projection and spiking activation generate the output. This design preserves both temporal causality and spike-driven sparsity. The full procedure is summarized in Appendix A.2.

### 3.2.3 SPIKE FEED-FORWARD NETWORK (SFFN)

The **Spike Feed-Forward Network (SFFN)** module follows the standard Transformer FFN structure but replaces activation functions with spiking neurons. Specifically, we support both the classic LIF neuron and the ternary spiking neuron introduced in SpikingLLM-v2. The module is defined as:

$$\mathcal{FC}(x) = \text{SpikeNeuron}(\mathcal{W}x + b), \tag{5}$$

$$\text{SFFN}(x) = \mathcal{FC}_2(\mathcal{FC}_1(x)), \tag{6}$$

where `SpikeNeuron` represents either a LIF or ternary spiking activation depending on the model variant. This formulation allows the feedforward block to remain fully spike-driven while supporting richer information encoding in SpikingLLM-v2.

### 3.3 SPIKE-FORM KNOWLEDGE DISTILLATION

To enable effective knowledge transfer from the teacher Artificial Neural Network (ANN) to the student Spiking Neural Network (SNN), we propose a novel framework called **Spike-Form Knowledge distillation (SKD)**. It consists of five key components targeting different representational levels as shown in Figure 3. Given the potential structural mismatch between teacher and student (e.g., in embedding dimensions, number of layers, or attention heads), we introduce structural alignment techniques to ensure compatibility, such as linear projections for dimension matching, head-wise mapping or projection for attention alignment, and layer skipping to bridge differing network depths.

Among various alignments, we focus on **Spike-Attention Alignment** and **Spike-Feature Alignment**. The reason is that other alignments (embedding, soft/hard targets) are largely consistent

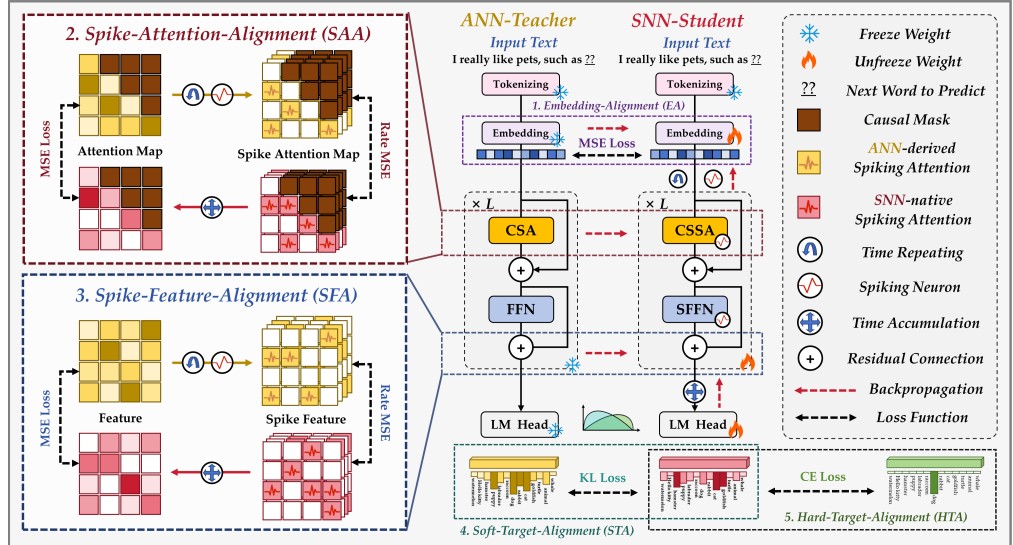

Figure 3: Overview of our **Spike-Form Knowledge Distillation (SKD)** framework. Knowledge is transferred from a frozen ANN teacher to a trainable SNN student via five alignment modules: (1) **Embedding Alignment** (EA); (2) **Spike-Attention Alignment** (SAA); F (3) **Spike-Feature Alignment** (SFA); (4) **Soft-Target Alignment** (STA); and (5) **Hard-Target Alignment** (HTA). Losses include MSE, CE, and spike-aware temporal strategies. In particular, our proposed *Rate-MSE* loss (equation 9) aligns the attention dynamics between ANN and SNN models over time. Dashed arrows indicate loss paths; spike-related operations are denoted with icons.

between ANNs and SNNs, while these two exhibit substantial differences: ANN representations are floating-point values, whereas SNN representations are discrete spikes (0-1), and they also include a temporal dimension. Detailed formulation and implementation of these alignments are provided in the Appendix A.3.

**Spike-Attention Alignment**  Given the fundamental difference in attention mechanisms, floating-point representations in the ANN versus spike-based representations in the SNN, as well as the additional temporal dimension in the SNN's attention outputs, which causes dimensional mismatch, we design two alignment strategies to enable effective cross-domain knowledge transfer:

**(a) Temporal Replication and Spiking:** We replicate the static attention map $A_{\text{ANN}} \in \mathbb{R}^{L \times L}$ across $T$ time steps:

$$\tilde{A}_{\text{ANN}} = \text{Repeat}(A_{\text{ANN}}, T) \in \mathbb{R}^{T \times L \times L}. \tag{7}$$

Each time step is then passed through a spiking neuron:

$$\hat{A}_{\text{spike}}^{\text{ANN}}(t) = \sigma_{\text{spike}}(\tilde{A}_{\text{ANN}}(t)), \quad t = 1, \dots, T. \tag{8}$$

We compute Rate-MSE loss:

$$\mathcal{L}_{\text{attn}}^{\text{RateMSE}} = \text{MSE}\left(\frac{1}{T}\sum_t \hat{A}_{\text{spike}}^{\text{ANN}}(t), \frac{1}{T}\sum_t A_{\text{SNN}}(t)\right). \tag{9}$$

**(b) Temporal Fusion and Distribution Matching:** Alternatively, we temporally average the SNN spike-attention and match it to the ANN attention using Mean Squared Error (MSE):

$$\bar{A}_{\text{SNN}} = \frac{1}{T}\sum_{t=1}^{T} A_{\text{SNN}}(t), \quad \mathcal{L}_{\text{attn}}^{\text{MSE}} = \text{MSE}(A_{\text{ANN}}, \bar{A}_{\text{SNN}}). \tag{10}$$

The overall spike-attention alignment loss is:

$$\mathcal{L}_{\text{attn}} = \alpha_1 \mathcal{L}_{\text{attn}}^{\text{RateMSE}} + \alpha_2 \mathcal{L}_{\text{attn}}^{\text{MSE}}. \tag{11}$$

**Spike-Feature Alignment**  Similarly, intermediate hidden states are aligned using a combination of rate-based and temporally averaged MSE:

$$\mathcal{L}_{\text{feat}} = \beta_1 \mathcal{L}_{\text{feat}}^{\text{RateMSE}} + \beta_2 \mathcal{L}_{\text{feat}}^{\text{MSE}}, \tag{12}$$

with linear projections and skip-layer connections used to handle mismatched dimensions and depths.

**Total Loss**  Combining embedding alignment, spike-based alignments, and traditional distillation, the student SNN is supervised with the overall training objective:

$$\mathcal{L}_{\text{total}} = \lambda_1 \mathcal{L}_{\text{emb}} + \lambda_2 \mathcal{L}_{\text{attn}} + \lambda_3 \mathcal{L}_{\text{feat}} + \lambda_4 \mathcal{L}_{\text{soft}} + \lambda_5 \mathcal{L}_{\text{hard}}. \tag{13}$$

## 4 EXPERIMENTS

### 4.1 TRAINING DETAILS

We use **FineWeb-Edu** (Penedo et al., 2024), a high-quality subset of the FineWeb corpus curated for factual and educational content. A 10B-token portion of the dataset is selected for pretraining. Notably, our SpikingLLM models achieve competitive performance under strict energy constraints, despite being trained on orders-of-magnitude fewer tokens (1–10 billion) compared to conventional ANN counterparts (typically requiring more than 100 Billion tokens), even at reduced parameter scales (0.125B-1.3B). The detailed training setup are provided in the Appendix A.4.

Table 1: Comparison of performance and estimated energy efficiency between SpikingLLM and conventional ANN baselines on the ACC benchmark. SpikingLLM-v1 adopts classic LIF neurons (see equation 2) implemented via SpikingJelly, while SpikingLLM-v2 employs ternary-valued spiking neurons with amplitude encoding (see equation 3), following the SpikeLM design. All energy estimates are calculated under a uniform FP32-based energy model for fair comparison. Time Step indicates the number of discrete simulation steps used during SNN inference.

| Model | Params (B) | Tokens (B) | Spike Form | Time Step | OPs (G) | Firing Rate | Energy (mJ) | Zero - shot Accuracy (%) ↑ | | | | | | | | |
|---|---|---|---|---|---|---|---|---|---|---|---|---|---|---|---|---|
| | | | | | | | | ARC-e | ARC-c | WG | BQ | PIQA | HS | OBQA | HQA | Avg. |
| OPT | 0.125 | 180 | × | — | 125.6 | — | 125.95 | 43.6 | 19.3 | 52.3 | 54.6 | 62.4 | 32.1 | 20.2 | 23.7 | 38.60 |
| Pythia | 0.160 | 300 | × | — | 125.7 | — | 126.01 | 43.7 | 19.8 | 52.8 | 55.1 | 62.7 | 33.6 | 20.1 | 24.2 | 39.00 |
| SpikeGPT | 0.046 | 16.5 | Binary | — | 3.66 | 0.174 | 3.29 | 32.3 | 16.2 | 50.2 | 45.7 | 54.6 | 25.3 | 15.7 | 20.6 | 32.58 |
| SpikeGPT | 0.216 | 16.5 | Binary | — | 18.3 | 0.168 | 16.53 | 35.2 | 17.7 | 50.7 | 47.3 | 55.1 | 27.6 | 17.3 | 23.1 | 34.25 |
| SpikingLLM-v1 | 0.125 | **1.0** | Binary | 2 | 12.1 | 0.196 | **5.24** | 39.1 | 18.9 | 50.3 | 52.7 | 56.7 | 28.1 | 19.8 | 22.9 | 36.05 |
| SpikingLLM-v2 | 0.125 | **1.0** | Ternary | 2 | 13.7 | 0.412 | **10.74** | 38.5 | 18.3 | 51.3 | 52.3 | 57.7 | 29.1 | 19.2 | 22.5 | 36.11 |
| SpikingLLM-v1 | 0.125 | **1.0** | Binary | 4 | 23.1 | 0.173 | **9.43** | 39.4 | 19.0 | 51.2 | 53.0 | 57.5 | 29.2 | 19.7 | 23.1 | 36.50 |
| SpikingLLM-v2 | 0.125 | **1.0** | Ternary | 4 | 25.8 | 0.386 | **19.92** | 38.9 | 18.5 | 51.5 | 52.9 | 58.0 | 28.3 | 19.2 | 22.9 | 36.27 |
| OPT | 0.350 | 180 | × | — | 360.8 | — | 197.57 | 47.5 | 22.2 | 55.3 | 57.2 | 66.1 | 40.7 | 25.7 | 26.6 | 42.68 |
| Pythia | 0.410 | 300 | × | — | 360.9 | — | 197.71 | 48.7 | 24.8 | 56.8 | 58.1 | 66.7 | 41.6 | 26.1 | 26.2 | 43.63 |
| SpikingLLM-v1 | 0.350 | **2.0** | Binary | 2 | 43.4 | 0.182 | **9.31** | 41.5 | 21.7 | 52.3 | 55.1 | 59.7 | 32.7 | 21.2 | 23.8 | 38.48 |
| SpikingLLM-v2 | 0.350 | **2.0** | Ternary | 2 | 47.7 | 0.404 | **18.61** | 41.5 | 21.7 | 52.4 | 54.9 | 59.1 | 31.6 | 20.8 | 23.1 | 38.14 |
| SpikingLLM-v1 | 0.350 | **2.0** | Binary | 4 | 84.2 | 0.178 | **16.75** | 42.1 | 21.4 | 52.1 | 56.1 | 60.5 | 33.1 | 21.9 | 23.5 | 38.84 |
| SpikingLLM-v2 | 0.350 | **2.0** | Ternary | 4 | 88.3 | 0.377 | **35.35** | 41.8 | 21.9 | 52.8 | 55.7 | 60.4 | 34.0 | 21.3 | 23.1 | 38.87 |
| OPT | 1.300 | 180 | × | — | 1237.1 | — | 632.22 | 57.8 | 30.4 | 60.4 | 60.8 | 71.7 | 52.6 | 33.4 | 30.7 | 49.73 |
| Pythia | 1.400 | 300 | × | — | 1237.4 | — | 632.48 | 60.5 | 31.2 | 61.3 | 61.1 | 71.1 | 53.6 | 33.2 | 31.9 | 50.49 |
| SpikingLLM-v1 | 1.300 | **10.0** | Binary | 2 | 66.7 | 0.192 | **37.16** | 45.7 | 23.5 | 54.2 | 56.3 | 62.3 | 40.2 | 24.5 | 24.0 | 41.33 |
| SpikingLLM-v2 | 1.300 | **10.0** | Ternary | 2 | 74.2 | 0.426 | **74.32** | 44.5 | 23.7 | 54.2 | 55.3 | 62.3 | 40.4 | 24.6 | 23.6 | 41.08 |
| SpikingLLM-v1 | 1.300 | **10.0** | Binary | 4 | 131.9 | 0.184 | **66.89** | 46.3 | 24.3 | 55.6 | 56.8 | 63.4 | 41.7 | 25.2 | 24.3 | 42.19 |
| SpikingLLM-v2 | 1.300 | **10.0** | Ternary | 4 | 141.6 | 0.411 | **141.21** | 45.8 | 24.5 | 55.6 | 56.1 | 63.4 | 41.3 | 25.5 | 24.8 | 42.12 |

### 4.2 MODEL EVALUATION

We evaluate models using zero-shot accuracy on diverse commonsense reasoning and QA benchmarks, including ARC-Easy (ARC-e), ARC-Challenge (ARC-c) (Clark et al., 2018), Winogrande (WG) (Sakaguchi et al., 2021), BoolQ (BQ) (Clark et al., 2019), PIQA (Bisk et al., 2020), HellaSwag (HS) (Zellers et al., 2019), OpenBookQA (OBQA) (Mihaylov et al., 2018), and HeadQA (HQA) (Vilares & Gómez-Rodríguez, 2019), measuring the generalization and reasoning abilities

without task-specific finetuning. As shown in Table 1, SpikingLLM achieves 82.60–94.56% of the zero-shot accuracy of counterpart ANN models at the same scale, despite using significantly fewer operations and training tokens. For example, SpikingLLM-v1 (1.3B, 10B tokens, 4 steps) reaches 42.19% accuracy versus 49.73% for OPT-1.3B, consuming only 10.6% of the energy per inference. For fair comparison with existing spiking LLMs, we focus on Zhu et al. (2023), a decoder-only SNN trained from scratch and architecturally comparable. We don't directly compare with Xing et al. (2024b) or Lv et al. (2023), which are not decoder-only and target different downstream tasks. And since Xing et al. (2024a) is derived via quantization and spiking conversion from pretrained ANN LLMs, we defer detailed comparisons with such quantization-based approaches to Appendix A.7.

### 4.3 ENERGY CONSUMPTION

To assess the efficiency of SNNs, we first measure the firing rate, defined as the average proportion of active spikes, where lower rates indicate higher sparsity and greater energy efficiency. Based on the firing rate, we then estimate the theoretical energy consumption during inference by simulating a 45nm neuromorphic chip, following Horowitz (2014); Kundu et al. (2021a); Yin et al. (2021); Kim & Panda (2021). Energy estimates are based on the total number of spike operations (SOPs), compared against floating-point operations (FLOPs) in baseline ANN models. Detailed computation steps are provided in Appendix A.6. The Table 1 reports the per-sample energy consumption, firing rates, and zero-shot accuracy across benchmarks. Our results show that: SpikingLLM-v1 consistently consumes an order of magnitude less energy than ANN baselines (e.g., 9.43 mJ vs. 126.01 mJ at 125M) while achieving over 93% of the accuracy. Across parameter scales (0.125B–1.3B), SpikingLLM maintains competitive performance at only **4.16%–5.87%** of the computational cost. Moreover, increasing time steps slightly improves performance (36.05% $\rightarrow$ 36.50% at 125M) with moderate energy overhead. SpikingLLM-v2 offers slightly higher accuracy at increased energy, providing a flexible trade-off for application constraints. These findings validate the viability of SNN-based LLMs for energy-constrained environments, such as edge devices and neuromorphic accelerators.

### 4.4 ABLATION STUDY

We conduct a series of ablation experiments on the **SpikingLLM-v1** model with 125M parameters to evaluate the contributions of key components and training factors. Specifically, we investigate: (1) the role of spike-driven modules, (2) the impact of varying simulation time steps, (3) the influence of training token volume, and (4) the effectiveness of our multi-level distillation alignment strategy.

**Spike-driven Modules** We first examine the effect of spike-driven modules by replacing CSSA and SFFN with their ANN counterparts. As shown in Figure 4a, the fully spike-driven design (CSSA+SFFN) achieves 36.05% accuracy with only 5.24 mJ energy. Replacing either module slightly improves accuracy (up to 36.57%) but increases energy consumption by more than $10\times$. Using ANN attention and FFN together yields 37.10% accuracy at the cost of $24\times$ higher energy. These results highlight that CSSA and SFFN are essential for preserving the energy-efficiency advantage of SpikingLLM.

**Distillation Alignment Strategy** We assess each alignment component by progressively adding it to the base HTA model. As shown in Figure 4b, STA yields the largest individual gain (+1.11%), while EA offers a smaller effect (+0.33%)., suggesting limited standalone benefit of energy alignment at this stage. Higher-level constraints such as SFA and SAA further improve performance (+0.63% and +0.38%). When combining STA with EA or SFA, the improvements increase more significantly, indicating complementary effects. Combining multiple objectives produces stronger gains, and the full set (STA, EA, SFA, SAA) achieves the best accuracy (36.25%, +1.68%). These results highlight the complementary benefits of hierarchical alignment for effective knowledge transfer from ANN teachers to spiking students.

**Time Steps** We further study the effect of varying simulation time steps (1–8). As shown in Figure 4c, more time steps improve accuracy by refining temporal resolution, but gains saturate beyond 4 steps while energy cost rises sharply. Firing rates gradually decline with longer steps,

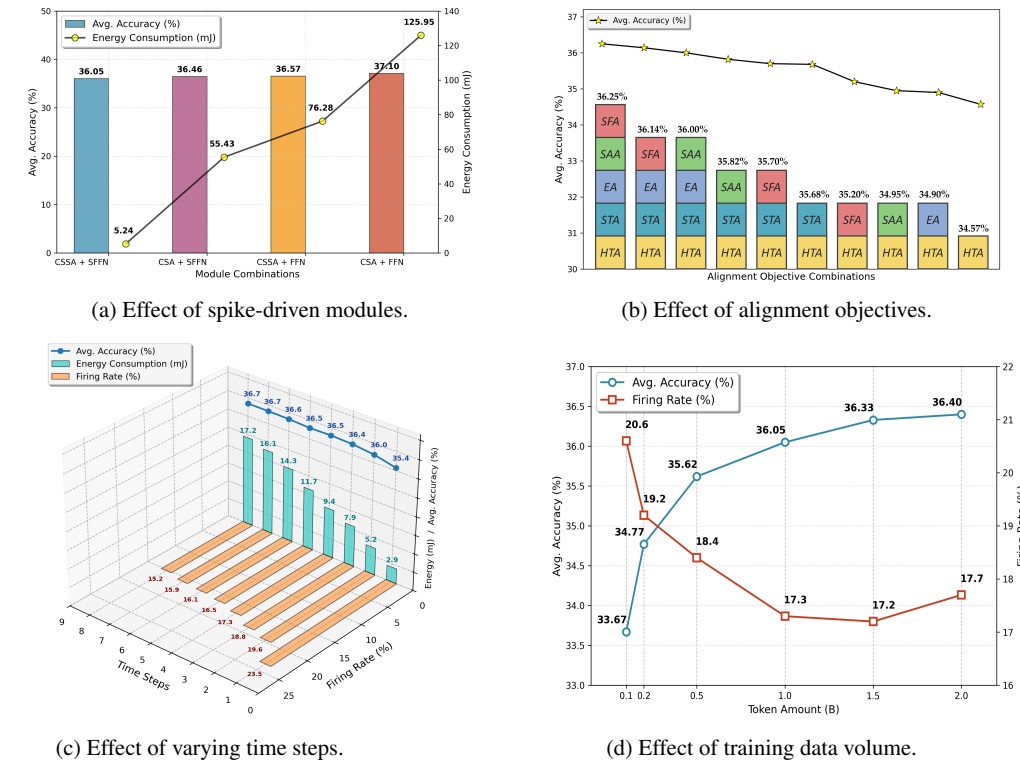

(a) Effect of spike-driven modules.  (b) Effect of alignment objectives.

(c) Effect of varying time steps.  (d) Effect of training data volume.

Figure 4: Visualization of ablation experiments.

indicating increased sparsity. Overall, 2–4 steps provide a good trade-off between efficiency and performance, while more steps yield marginal accuracy gains at higher energy cost.

**Training Token Volume** Finally, we evaluate the impact of training data size by varying tokens from 0.1B to 2.0B. As shown in Figure 4d, accuracy improves consistently with more data, with the largest gains in the low-data regime (0.1B → 0.5B) and saturation beyond 1.0B tokens. Performance rises from 33.67% to 36.40%, reaching 94.3% of the teacher's accuracy (38.60%). Interestingly, the firing rate gradually decreases as training data increases, suggesting that larger training dataset not only improves performance but also enhances temporal sparsity, likely due to more structured representations. These results demonstrate the data efficiency of our training framework, enabling near-saturated performance with relatively few tokens. Further details on firing and activation patterns are visualized in the Appendix A.8.

## 5 CONCLUSION

We introduce **SpikingLLM**, the fully binary spike-driven LLM trained from random initialization. Its **Causal Spiking Self-Attention (CSSA)** enables softmax-free, spike-based autoregressive modeling, reducing computational cost by over $10\times$ compared to ANNs. A multi-level **Spike-Form Knowledge Distillation (SKD)** framework further improves performance by aligning representations across multiple levels. SpikingLLM achieves competitive accuracy with fewer training tokens and lower energy, demonstrating a promising pathway for energy-efficient, brain-inspired NLP.

**Limitations:** While SpikingLLM significantly reduces computational cost, its accuracy still lags behind large-scale ANN LLMs on some benchmarks, and training larger models requires careful tuning of time steps and distillation schedules. Future work could explore improved spike-based architectures and more effective distillation strategies to further close the gap with ANN performance.

## ETHICS STATEMENT.

All experiments in this work are conducted on publicly available datasets without involving private or sensitive information. The proposed methods are intended purely for academic research, and any deployment should carefully consider potential ethical risks such as bias or misuse.

## REPRODUCIBILITY STATEMENT.

The experimental results in this paper are reproducible. We describe the model architecture and training process details in the main text and appendix. We will release the source code after review.

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

# A  APPENDIX

## A.1  USE OF LLMS

In this work, we used Large Language Models (LLMs) in a limited and auxiliary capacity. Specifically, LLMs were employed for retrieval and discovery of related literature on Spiking Neural Networks (SNNs), neuromorphic computing, and energy-efficient large language models. This assisted us in identifying relevant prior work and ensuring broader coverage of existing approaches. Importantly, LLMs were not involved in designing algorithms, implementing models, or analyzing experimental results. All methodological innovations, including the Causal Spiking Self-Attention (CSSA) and Spike-Form Knowledge Distillation (SKD), were independently conceived, implemented, and validated by the authors. Thus, the role of LLMs was restricted to accelerating literature exploration, without influencing the scientific contributions of this paper.

## A.2  ALGORITHM PROCEDURE OF CSSA

---

**Algorithm 1** Causal Spiking Self-Attention (CSSA)

---

**Input**: Spike-based input $X$
**Output**: Spiking attention output

1: **// Step 1: Input Projection** ($FP \leftarrow Spike @ FP$)
2: $q, k, v \leftarrow \text{Linear}_{Q,K,V}(X)$
3: **// Step 2: Spiking Neuron Encoding** ($Spike \leftarrow FP$)
4: $\text{spike}_q \leftarrow SpikingNeuron_Q(q)$
5: $\text{spike}_k \leftarrow SpikingNeuron_K(k)$
6: $\text{spike}_v \leftarrow SpikingNeuron_V(v)$
7: **// Step 3: Attention** ($Integer \leftarrow Spike @ Spike$)
8: $\text{attn\_int} \leftarrow \text{spike}_q @ \text{spike}_k^T$
9: **// Step 4: Causal Masking and Spiking**
10: $\text{causal\_mask} \leftarrow \text{causal\_mask} \odot \text{attn\_mask}$
11: $\text{attn\_causal} \leftarrow \text{causal\_mask} \odot \text{attn\_int}$
12: $spike_{attn} \leftarrow SpikingNeuron_{Attn}(\text{attn\_causal})$
13: **// Step 5: Summation** ($Integer \leftarrow Spike @ Spike$)
14: $\text{attn\_out} \leftarrow spike_{attn} @ \text{spike}_v^T$
15: $spike_{attn\_out} \leftarrow SpikingNeuron_{AttnOut}(\text{attn\_out})$
16: **// Step 6: Output Projection** ($FP \leftarrow Spike @ FP$)
17: $\text{fp\_out} \leftarrow \text{Linear}_{\text{out}}(spike_{attn\_out})$
18: **// Step 7: Spiking** ($Spike \leftarrow FP$)
19: $\text{spike\_out} \leftarrow SpikingNeuron_{Out}(\text{fp\_out})$
20: **return** spike_out

---

## A.3  DETAILED DESIGN OF SKD

We present Spike-Form Knowledge Distillation (SKD), a framework that distills a frozen ANN teacher into a trainable SNN student. Distillation proceeds through five aligned losses: Embedding Alignment (EA), Soft-Target Alignment (STA), and Hard-Target Alignment (HTA) reuse standard MSE/CE because continuous vectors are already compatible; Spike-Attention Alignment (SAA) and Spike-Feature Alignment (SFA) introduce spike-aware temporal losses to bridge the unique continuous-to-binary and spatial-to-temporal gap that only these two representations expose.

### A.3.1  EMBEDDING ALIGNMENT

We align the output distributions from the embedding layers of the teacher and student networks using Mean Squared Error (MSE) loss:

$$\mathcal{L}_{\text{emb}} = \text{MSE}(P_{\text{emb}}^{\text{ANN}}, P_{\text{emb}}^{\text{SNN}}). \tag{14}$$

This alignment ensures consistent semantic representations at the input level, easing the optimization burden and improving representation consistency across modalities. A linear transformation is applied if the embedding dimensions are not directly compatible.

### A.3.2 SPIKE-ATTENTION ALIGNMENT

Given the fundamental difference in attention mechanisms—floating-point representations in the ANN versus spike-based representations in the SNN—as well as the additional temporal dimension in the SNN's attention outputs, which causes dimensional mismatch, we design two alignment strategies to enable effective cross-domain knowledge transfer:

**(a) Temporal Replication and Spiking:** We replicate the static attention map $A_{\mathrm{ANN}} \in \mathbb{R}^{L \times L}$ across $T$ time steps:

$$\tilde{A}_{\mathrm{ANN}} = \mathrm{Repeat}(A_{\mathrm{ANN}}, T) \in \mathbb{R}^{T \times L \times L}. \tag{15}$$

Each time step is then passed through a spiking neuron:

$$\hat{A}_{\mathrm{spike}}^{\mathrm{ANN}}(t) = \sigma_{\mathrm{spike}}(\tilde{A}_{\mathrm{ANN}}(t)), \quad t = 1, \ldots, T. \tag{16}$$

We compute Rate-MSE loss:

$$\mathcal{L}_{\mathrm{attn}}^{\mathrm{RateMSE}} = \mathrm{MSE}\left( \frac{1}{T} \sum_t \hat{A}_{\mathrm{spike}}^{\mathrm{ANN}}(t), \frac{1}{T} \sum_t A_{\mathrm{SNN}}(t) \right). \tag{17}$$

**(b) Temporal Fusion and Distribution Matching:** Alternatively, we temporally average the SNN spike-attention and match it to the ANN attention using Mean Squared Error (MSE):

$$\bar{A}_{\mathrm{SNN}} = \frac{1}{T} \sum_{t=1}^{T} A_{\mathrm{SNN}}(t), \quad \mathcal{L}_{\mathrm{attn}}^{\mathrm{MSE}} = \mathrm{MSE}(A_{\mathrm{ANN}}, \bar{A}_{\mathrm{SNN}}). \tag{18}$$

The overall spike-attention alignment loss is:

$$\mathcal{L}_{\mathrm{attn}} = \alpha_1 \mathcal{L}_{\mathrm{attn}}^{\mathrm{RateMSE}} + \alpha_2 \mathcal{L}_{\mathrm{attn}}^{\mathrm{MSE}}. \tag{19}$$

### A.3.3 SPIKE-FEATURE ALIGNMENT

To align intermediate hidden states, we apply the same transformation strategies to the feature maps:

$$\mathcal{L}_{\mathrm{feat}} = \beta_1 \mathcal{L}_{\mathrm{feat}}^{\mathrm{RateMSE}} + \beta_2 \mathcal{L}_{\mathrm{feat}}^{\mathrm{MSE}}, \tag{20}$$

where each component is computed similarly to the attention alignment, but on the feature tensors $H_{\mathrm{ANN}}$ and $H_{\mathrm{SNN}}$. And linear projections are inserted if the hidden dimensions differ. Skip-layer connections are used if the number of layers does not match.

### A.3.4 SOFT TARGET ALIGNMENT

We apply soft-label distillation using the teacher and student logits:

$$\mathcal{L}_{\mathrm{soft}} = \mathrm{KL}\left( \frac{\mathrm{logits}_{\mathrm{ANN}}}{\tau} \,\middle\|\, \frac{\mathrm{logits}_{\mathrm{SNN}}}{\tau} \right), \tag{21}$$

where $\tau$ is a temperature hyperparameter to soften the logits.

### A.3.5 HARD TARGET ALIGNMENT

We also include the traditional cross-entropy loss with the ground truth:

$$\mathcal{L}_{\mathrm{hard}} = \mathrm{CE}(\mathrm{logits}_{\mathrm{SNN}}, y). \tag{22}$$

### A.3.6 TOTAL LOSS

The final training objective combines all loss terms:

$$\mathcal{L}_{\text{total}} = \lambda_1 \mathcal{L}_{\text{emb}} + \lambda_2 \mathcal{L}_{\text{attn}} + \lambda_3 \mathcal{L}_{\text{feat}} + \lambda_4 \mathcal{L}_{\text{soft}} + \lambda_5 \mathcal{L}_{\text{hard}}. \tag{23}$$

Each $\lambda_i$ balances the contribution of its corresponding component.

### A.4 TRAINING DETAILS

Table 2: Summary of training hyperparameters and configurations used for SpikingLLM, including optimization settings, distillation parameters, and hardware specifications.

| Hyperparameter | Value |
|---|---|
| Teacher ANN model | OPT-family |
| Student SNN model | SpikingLLM |
| Tokenizer / Vocabulary | Aligned with OPT |
| Batch size | 16 |
| Gradient accumulation steps | 16 |
| Effective batch size | 256 |
| Optimizer | Adam |
| Learning rate | $5 \times 10^{-4}$ |
| Scheduler | Cosine decay |
| Warm-up ratio | 0.2 |
| Gradient clipping threshold | 0.7 |
| Temperature $\tau$ (for SKD) | 2.0 |
| Distillation weights ($\lambda_1$ to $\lambda_5$) | 0.2, 0.1, 0.1, 0.3, 0.3 |
| Inference time steps ($T$) | 2 and 4 |
| Hardware | NVIDIA RTX 4090 (24GB) |

**Training Paradigm**   The training of **SpikingLLM** follows a teacher–student paradigm, where the teacher model is a pre-trained open-source ANN-based large language model from the OPT family, and the student is our spike-based SpikingLLM. To ensure consistency between the teacher and student models, we align both the vocabulary and tokenizer with those used in OPT.

**Optimization Setup**   All experiments are conducted using a batch size of 16 and a gradient accumulation factor of 16, effectively yielding a total batch size of 256 tokens. The optimizer used is Adam, with a learning rate set to $5 \times 10^{-4}$. A cosine learning rate scheduler with a warm-up ratio of 0.2 is employed to stabilize early training. Gradient clipping is applied with a threshold of 0.7 to avoid exploding gradients in the early training phases, which can be particularly pronounced in spiking models. And all models are trained on NVIDIA 4090 GPUs with 24GB memory.

**Spike-Form Knowledge Distillation**   For spike-form knowledge distillation (SKD), we adopt a temperature of $\tau = 2.0$ in the soft targets from the teacher model. The overall loss is computed as a weighted combination of multiple alignment objectives defined in Method Section. The corresponding loss weights are: $\lambda_1 = 0.2$, $\lambda_2 = 0.1$, $\lambda_3 = 0.1$, $\lambda_4 = 0.3$, and $\lambda_5 = 0.3$.

**Inference Time Steps**   To study the trade-off between accuracy and energy efficiency, SpikingLLM is trained with varying numbers of inference time steps, specifically $T = 2$ and $T = 4$. A higher number of steps improves temporal resolution and accuracy at the cost of increased energy consumption, enabling flexible deployment depending on the application constraints.

**Training Data Selection**   The subset of training data used in our experiments was drawn from the fineweb-edu dataset (Penedo et al., 2024), specifically the 10BT sample accessible at `https://huggingface.co/datasets/HuggingFaceFW/fineweb-edu/tree/main/sample/10BT`. Since our claim of using a lower training token volume is a core contribution, it is crucial to detail how the data was sampled to ensure reproducibility. This subset was selected directly from the publicly available sample without additional filtering or preprocessing, providing other researchers with a clear and reproducible training set.

## A.5 SURROGATE GRADIENT

Training spiking neural networks (SNNs) presents a significant challenge due to the non-differentiable nature of spike generation functions, such as the Heaviside step function used in the spiking neuron model. To enable end-to-end optimization with backpropagation, we adopt a surrogate gradient approach introduced by Fang et al. (2020).

Specifically, the discrete spiking activation $S$ is approximated by a continuous and differentiable function using an arctangent-based surrogate:

$$S \approx \frac{1}{\pi} \arctan\left(\frac{\pi}{2}\alpha U\right) + \frac{1}{2}, \tag{24}$$

where $U$ is the membrane potential and $\alpha$ is a tunable hyperparameter controlling the sharpness of the transition. In our experiments, we set $\alpha = 2$ by default, balancing gradient magnitude and smoothness.

Taking the derivative of Equation equation 24 yields the surrogate gradient used during backpropagation:

$$\frac{\partial S}{\partial U} = \frac{\alpha}{2} \cdot \frac{1}{1 + \left(\frac{\pi}{2}\alpha U\right)^2}. \tag{25}$$

This surrogate formulation enables stable and effective gradient-based optimization for SpikingLLM. It allows error signals to be backpropagated through spike-generating layers without requiring exact gradients, thus making the training pipeline compatible with standard deep learning frameworks.

## A.6 THEORETICAL SYNAPTIC OPERATION AND ENERGY CONSUMPTION CALCULATION

The theoretical energy consumption of *SpikingLLM* is estimated by first calculating the synaptic operations (SOPs). For each block or layer $l$, we have:

$$\text{SOPs}(l) = f_r(l) \times T \times \text{FLOPs}(l), \tag{26}$$

where $l$ indexes a block in *SpikingLLM*, $f_r(l)$ is the average firing rate of the input spike train to block $l$ (measured as spikes per neuron per time step), and $T$ is the simulation time steps of the spiking neuron. FLOPs$(l)$ denotes the number of multiply-and-accumulate (MAC) operations of block $l$ in the equivalent ANN. SOPs$(l)$ thus represents the spike-based accumulate (AC) operations performed in the SNN.

Following Horowitz (2014), we assume the energy per operation on a 45 nm process as

$$E_{\text{MAC}} = 4.6\,\text{pJ}, \qquad E_{\text{AC}} = 0.9\,\text{pJ}.$$

For ANNs, the theoretical energy consumption of a block $b$ is

$$\text{Power}_{\text{ANN}}(b) = E_{\text{MAC}} \times \text{FLOPs}(b). \tag{27}$$

For SNNs, the theoretical energy consumption of block $b$ is

$$\text{Power}_{\text{SNN}}(b) = E_{\text{AC}} \times \text{SOPs}(b). \tag{28}$$

According to ( (Horowitz, 2014; Kundu et al., 2021a;b; Hu et al., 2021; Yin et al., 2021; Kim & Panda, 2021; Yao et al., 2021)), the total energy consumption of *SpikingLLM* can be decomposed into three parts: (1) the embedding stage, which is executed with dense MAC operations, (2) the $L$ stacked transformer blocks, each of which is spiking and therefore counted using AC operations, and (3) the language-model head (LM-head) that maps hidden states to vocabulary logits (dense MACs). We write:

$$E_{\text{SpikingLLM}} = E_{\text{MAC}} \cdot \left(\text{FLOPs}_{\text{Embed}} + \text{FLOPs}_{\text{LM-head}}\right)$$
$$+ E_{\text{AC}} \cdot \sum_{l=1}^{L} \left(\text{SOP}_{\text{CSSA}}(l) + \text{SOP}_{\text{SFFN}}(l)\right) \tag{29}$$

where FLOPs $_\text{Embed}$ and FLOPs $_\text{LM-head}$ denote the MAC operations of the embedding stage and the output projection to vocabulary logits, respectively; SOP$_\text{CSSA}(l)$ and SOP$_\text{SFFN}(l)$ represent the spike-accumulate operations of the Spiking Causal Self-Attention and Spiking Feed-Forward Network modules in block $l$; $E_\text{MAC}$ and $E_\text{AC}$ are the energy costs per MAC and AC operation; $L$ is the number of transformer blocks; and $f_r$ and $T$ denote the average firing rate and the number of simulation time steps.

## A.7    COMPARISON WITH SPIKELLM

Table 3: Comparison of SpikingLLM and SpikeLLM across different model scales, neuron/spike formats, and time steps. Avg. Acc. reports zero-shot accuracy (%), and SNN/ANN Ratio shows the performance of spiking models relative to their ANN counterparts.

| Model | Params (B) | Tokens (B) | Spike Form | Time Step | Avg. Acc.(%) ↑ | SNN/ANN Ratio (%) ↑ |
|---|---|---|---|---|---|---|
| SpikingLLM-v1 | 0.125 | **1.0** | Binary | 2 | 36.05 | 93.39 |
| SpikingLLM-v2 | 0.125 | **1.0** | Ternary | 2 | 36.11 | 93.55 |
| SpikingLLM-v1 | 0.125 | **1.0** | Binary | 4 | 36.50 | 94.56 |
| SpikingLLM-v2 | 0.125 | **1.0** | Ternary | 4 | 36.27 | 93.96 |
| SpikingLLM-v1 | 0.350 | **2.0** | Binary | 2 | 38.48 | 90.16 |
| SpikingLLM-v2 | 0.350 | **2.0** | Ternary | 2 | 38.14 | 89.36 |
| SpikingLLM-v1 | 0.350 | **2.0** | Binary | 4 | 38.84 | 91.00 |
| SpikingLLM-v2 | 0.350 | **2.0** | Ternary | 4 | 38.87 | 91.07 |
| SpikingLLM-v1 | 1.300 | **10.0** | Binary | 2 | 41.33 | 83.11 |
| SpikingLLM-v2 | 1.300 | **10.0** | Ternary | 2 | 41.08 | 82.60 |
| SpikingLLM-v1 | 1.300 | **10.0** | Binary | 4 | 42.19 | 84.84 |
| SpikingLLM-v2 | 1.300 | **10.0** | Ternary | 4 | 42.12 | 84.70 |
| SpikeLLM | 7.000 | — | Integer (W2A16) | 2 | 49.92 | 78.17 |
| SpikeLLM | 7.000 | — | Integer (W2A8) | 4 | 41.77 | 65.41 |
| SpikeLLM | 13.00 | — | Integer (W2A16) | 2 | 53.76 | 81.34 |
| SpikeLLM | 13.00 | — | Integer (W2A8) | 4 | 50.12 | 75.78 |
| SpikeLLM | 70.00 | — | Integer (W2A16) | 2 | 60.47 | 82.55 |

Table 3 compares SpikingLLM with SpikeLLM across different model scales, quantization methods, spike forms, and simulation time steps. Overall, SpikingLLM achieves competitive zero-shot accuracy with smaller models and lower-precision spike forms. Notably, the SNN/ANN ratio of SpikingLLM is consistently higher (82–95%) than that of SpikeLLM, indicating that our spike-based models retain more of the original ANN performance. This improvement is largely attributed to our multi-level knowledge distillation framework, which effectively transfers information from ANN teachers to spiking students. These results not only highlight the efficiency and effectiveness of our approach but also provide a promising pathway for further improving SNN-LLMs through advanced distillation strategies.

## A.8 FIRING VISUALIZATION

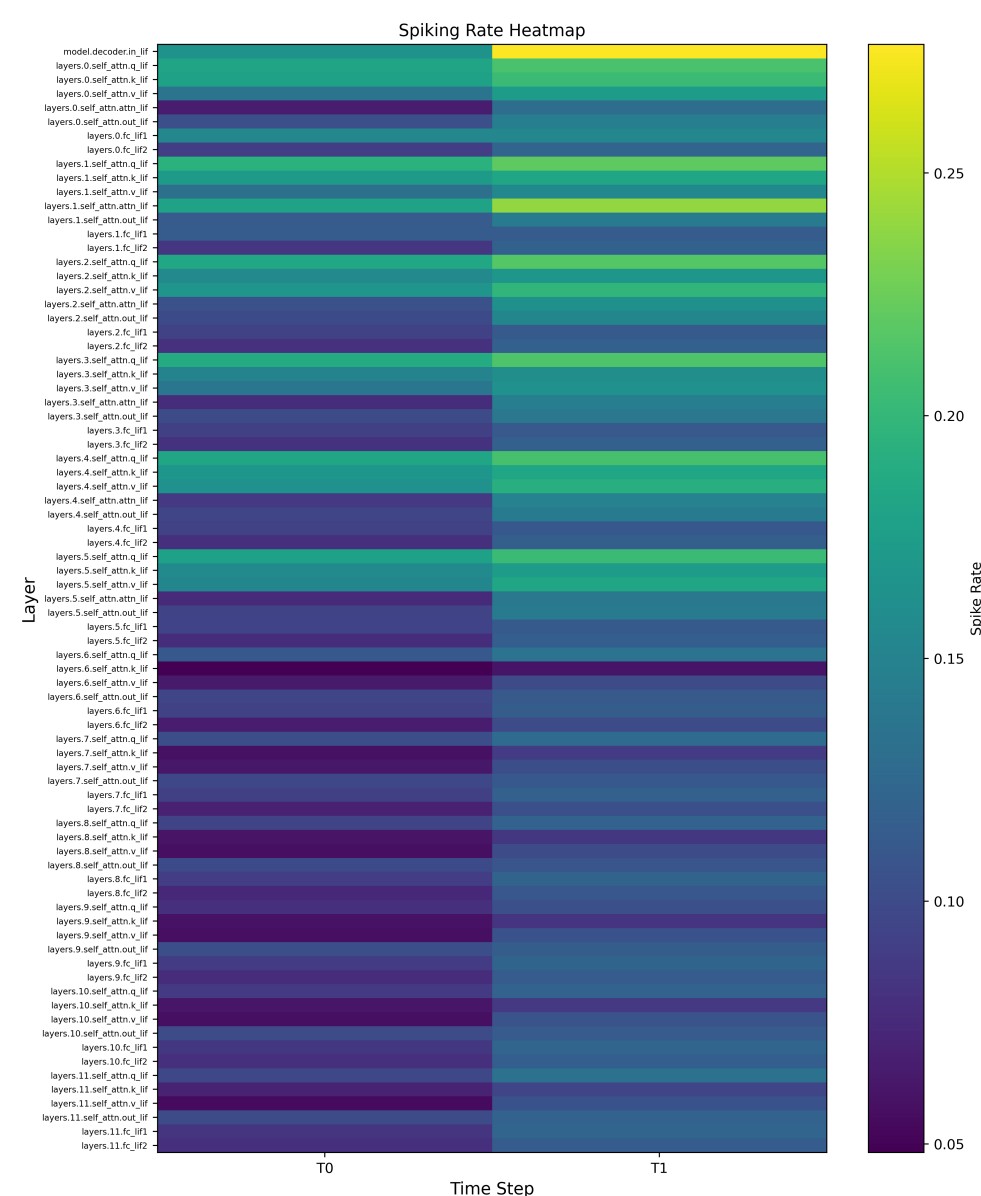

Figure 5: At $T = 2$, the firing activity is relatively concentrated in a few specific layers, as indicated by localized high-intensity regions in the heatmap. This suggests that, under limited temporal resolution, only a subset of layers become highly active, likely those responsible for early-stage processing and critical feature extraction. The rest of the network remains relatively quiescent, reflecting a sparse activation pattern constrained by the short integration window.

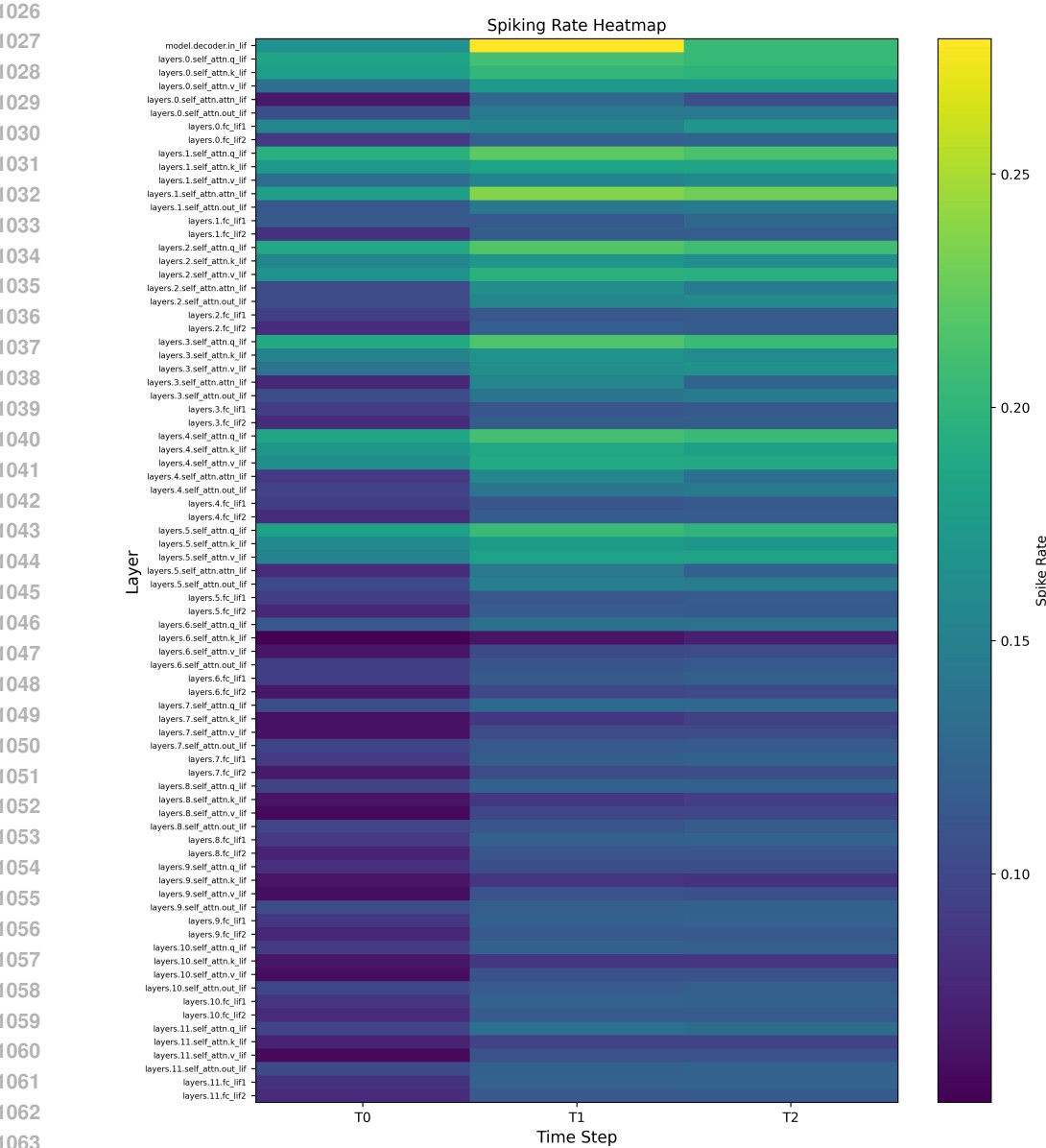

Figure 6: With the increase to three time steps, the regions of elevated firing rate begin to extend across more layers. This indicates that more layers participate in the computation as temporal resolution improves, enabling broader propagation of information. The increased coverage reflects a more distributed spiking pattern, suggesting enhanced temporal integration and coordination among layers.

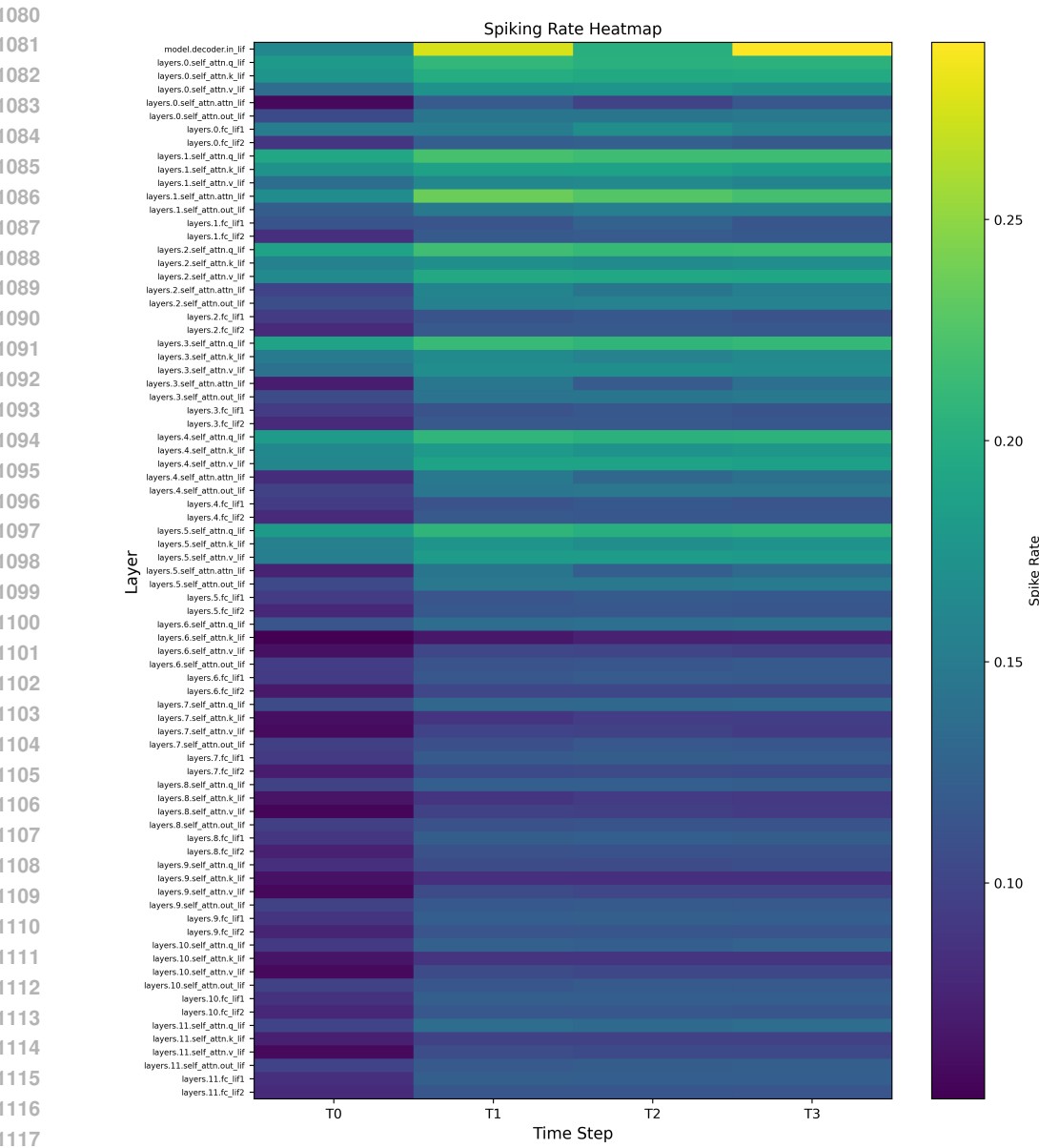

Figure 7: At $T = 4$, the firing activity becomes even more widespread, engaging a significant portion of the network. A greater number of layers exhibit moderate to high firing rates, which may reflect more comprehensive information processing and deeper hierarchical interactions. The broader engagement suggests that intermediate temporal budgets allow for more expressive internal representations.

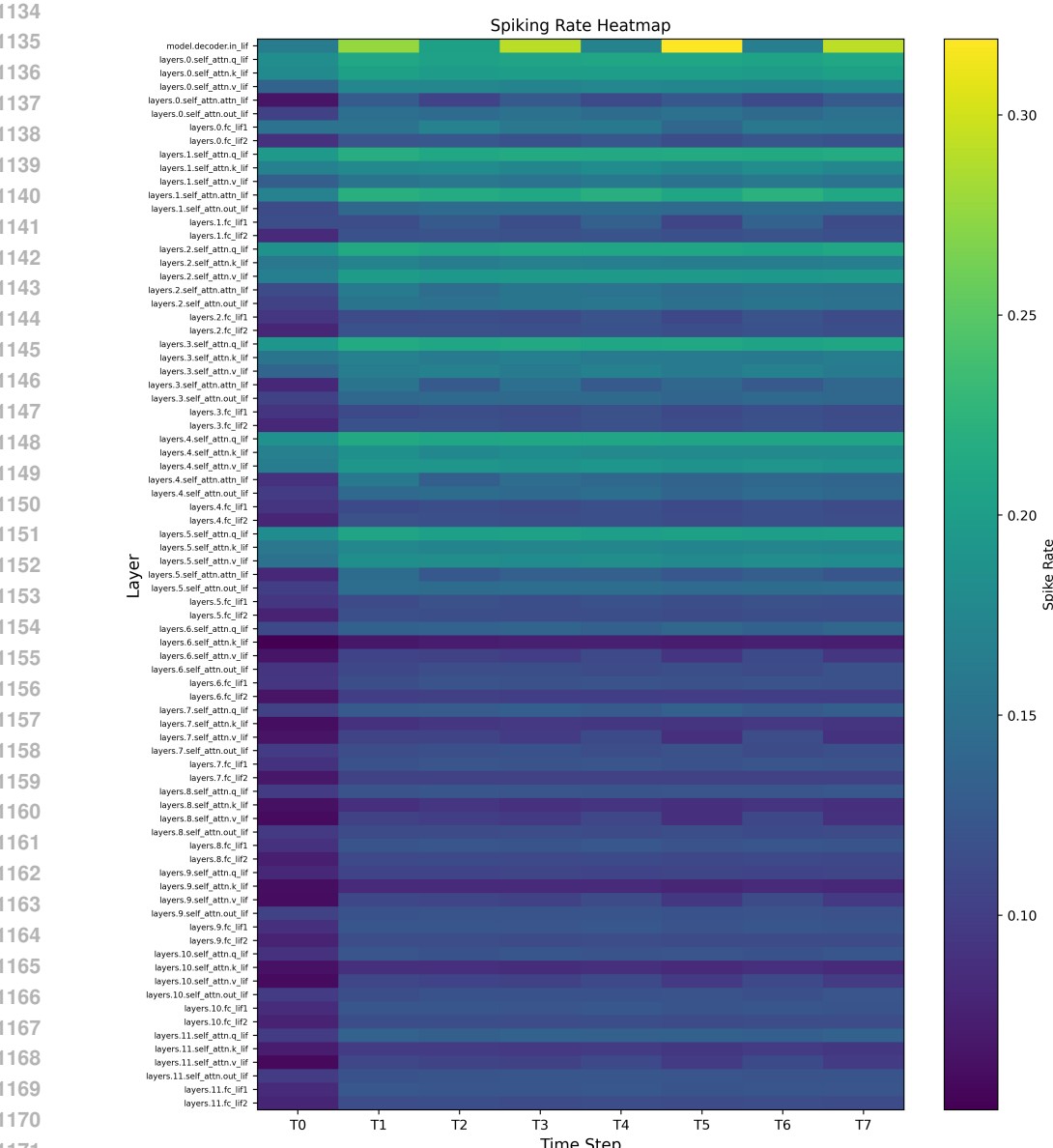

Figure 8: When the number of time steps is increased to $T = 8$, the firing distribution becomes relatively uniform across most layers. While some layers still exhibit elevated activity, the overall pattern is more homogeneous, indicating that nearly all layers participate in the computation to some extent. This may reflect a fully temporally saturated regime, where the network is capable of leveraging extended integration windows for more complex and nuanced feature extraction.

To investigate the temporal dynamics of spiking activity in **SpikingLLM-v1**, we visualize the firing rate distributions across layers under varying numbers of inference time steps: $T = 2$, $T = 3$, $T = 4$, and $T = 8$. The firing rate heatmaps are constructed such that the vertical axis corresponds to different model layers, the horizontal axis represents discrete time steps, and the color intensity encodes the normalized firing rate ranging from low (purple) to high (yellow) (see Figures 5–8).

Overall, the progression of firing rate distributions across increasing time steps reveals a transition from sparse and localized activation to distributed and pervasive spiking. This dynamic suggests that **SpikingLLM-v1** adapts its computational strategy based on the temporal budget: utilizing minimal resources under constrained settings (e.g., $T = 2$) and expanding activation as more time steps

become available. These results highlight the temporal adaptability of spiking neural architectures and their potential for scalable, energy-aware language processing.

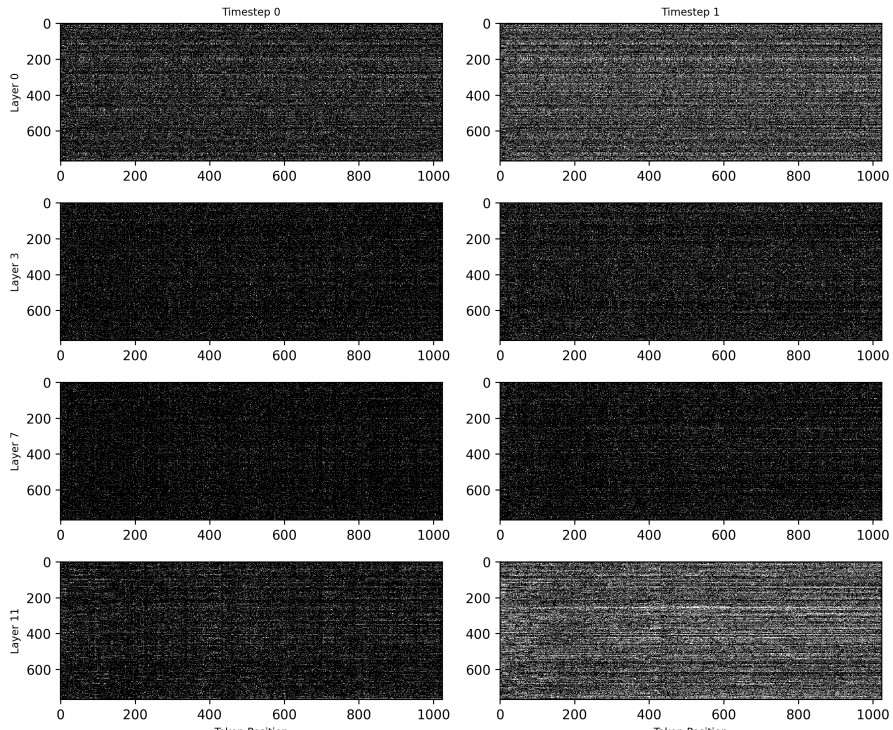

Figure 9: This figure presents the spiking activity across token positions for four representative layers (Layer 0, Layer 3, Layer 7, and Layer 11) at an early inference time step. The firing patterns are relatively sparse and uniformly distributed, particularly in the lower layers. This reflects the initial stage of neuronal processing, where the model begins encoding input signals with limited temporal context. Notably, deeper layers such as Layer 11 exhibit subdued activation, suggesting that higher-level abstractions have not yet emerged.

In addition to the layer-wise temporal spiking visualization, we further examine the firing patterns of **SpikingLLM-v1** at the level of individual token positions. A new set of visualizations (see Figures 9–12) illustrates the spiking activity across different token positions for selected layers (specifically, Layer 0, Layer 3, Layer 7, and Layer 11) under varying inference time steps. In these heatmaps, the vertical axis corresponds to token positions, the horizontal axis denotes discrete time steps, and the color intensity indicates the firing magnitude, with lighter colors representing stronger activity and darker regions indicating lower activation.

These token-level firing visualizations provide a more granular perspective on the internal computation dynamics of **SpikingLLM-v1**. The evolution of spiking activity across time steps reveals a clear progression: from diffuse and uniform firing in early layers and time steps, toward increasingly selective and structured activation in deeper layers as more time is allocated. This suggests a hierarchical processing mechanism wherein early layers operate in a temporally shallow regime, broadly encoding input stimuli, while deeper layers gradually accumulate temporal context to perform more abstract and task-specific computations. Overall, the model exhibits both spatial and temporal specialization, underscoring the potential of spiking neural architectures for dynamic and efficient information processing.

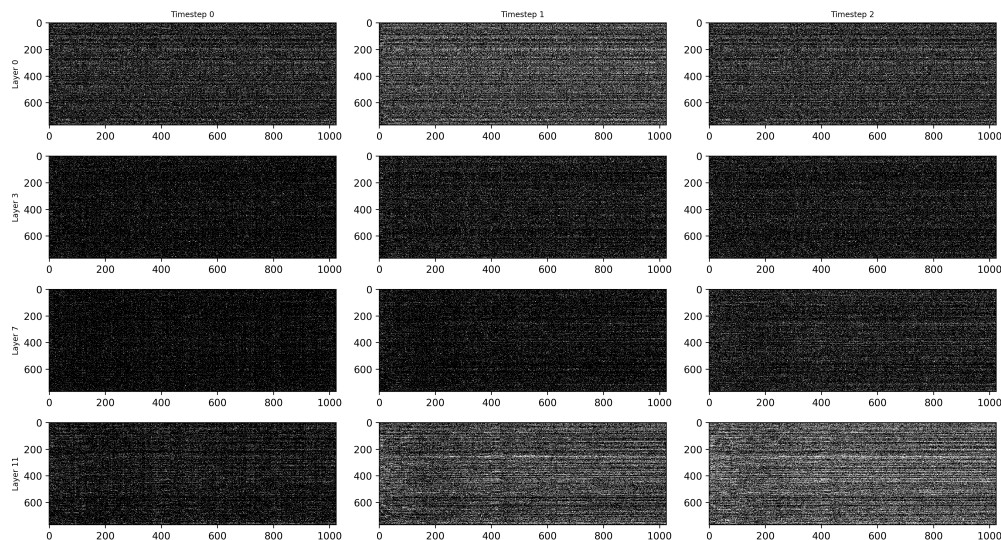

Figure 10: At $T = 3$, the firing distributions become slightly more structured across token positions and layers. While lower layers maintain broadly distributed activity, deeper layers begin to display early signs of selective activation. Compared to $T = 2$, this figure reveals the onset of temporal refinement, indicating that additional time steps allow the model to initiate more context-sensitive computation, particularly in the upper layers.

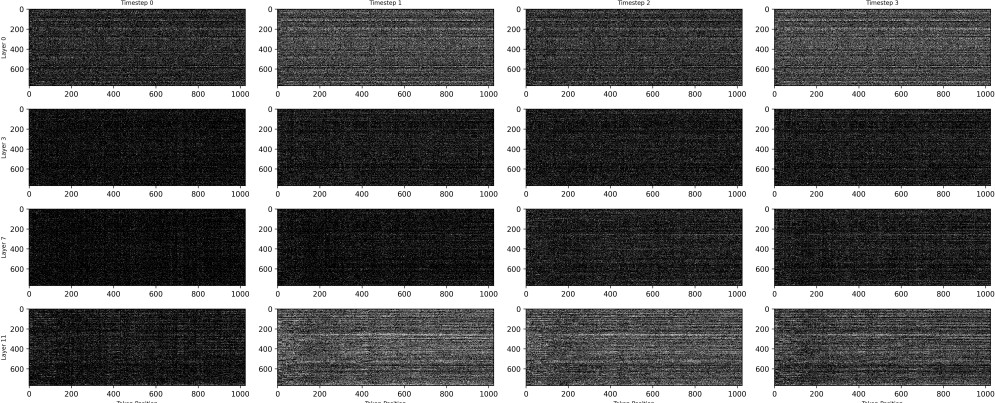

Figure 11: With four time steps, the model exhibits more pronounced spatiotemporal differentiation in firing behavior. Activity becomes more variable across token positions, and certain regions in deeper layers start to display concentrated firing. This suggests that the network is engaging in increasingly specialized processing, distributing its computation more selectively based on both input semantics and accumulated temporal evidence.

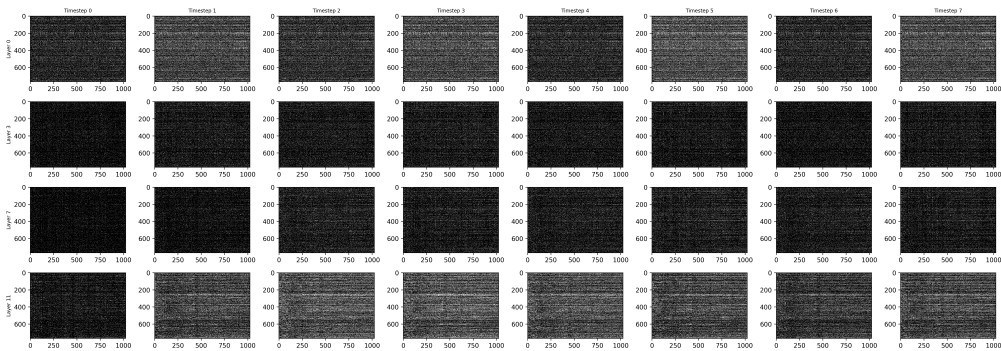

Figure 12: By $T = 8$, the firing patterns exhibit substantial temporal evolution and structural complexity. Deeper layers, in particular, show heightened and more focused activation for specific token regions, reflecting refined internal representations. This level of activity suggests that the model has transitioned into a more stable and semantically rich encoding phase. The marked increase in firing diversity and intensity across layers highlights the model's capacity to utilize extended temporal windows for deeper contextual integration and task-specific computation.

### A.9 WEIGHT VISUALIZATION

In this subsection, we analyze the weight distributions of Artificial Neural Networks (ANNs) and Spiking Neural Networks (SNNs) across different layers and components. The weight visualization provides valuable insights into the fundamental differences between these two types of neural networks and highlights the unique characteristics of SNNs. The weight distributions of ANNs and SNNs exhibit distinct patterns across various layers and components (q_proj, k_proj, v_proj, out_proj, fc1, fc2).

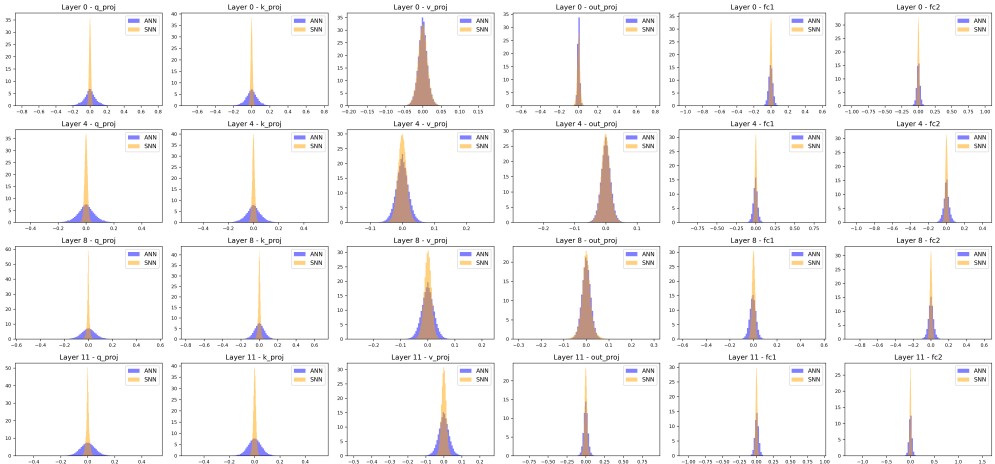

Figure 13: Weight distribution comparison between Artificial Neural Networks (ANNs) and Spiking Neural Networks (SNNs). ANNs typically exhibit a more concentrated weight distribution around zero, especially in early layers (e.g., Layer 0). In deeper layers (e.g., Layer 11), their weight distribution becomes slightly more spread out but remains relatively compact, indicating tightly clustered weights that contribute to stability and ease of training. In contrast, SNNs show a broader and more dispersed weight distribution, with weights less tightly clustered around zero. This broader spread is particularly notable in deeper layers (e.g., Layer 11), reflecting the dynamic and diverse weight updates characteristic of their spiking nature.

Our approach differs from traditional ANN-to-SNN conversion methods in that we do not passively fit SNN weights to match those of ANNs. Instead, we actively capture and adapt to the unique characteristics of SNNs. This active adaptation is crucial for leveraging the full potential of SNNs, which operate on spiking dynamics rather than continuous activation values. By focusing on the inherent properties of SNNs, such as their broader weight distribution and dynamic spiking behavior, our method ensures that the network is optimized for spiking neural computation. This approach allows SNNs to maintain their distinct advantages, such as energy efficiency and biological plausibility, while still achieving high performance.

In summary, the weight visualization clearly demonstrates the differences between ANNs and SNNs. Our method capitalizes on these differences by actively adapting to the unique characteristics of SNNs, rather than forcing them to conform to the weight and activation patterns of ANNs. This approach is essential for developing effective and efficient SNNs that can fully leverage their spiking dynamics.

### A.10 OUTLIER ANALYSIS OF WEIGHT DISTRIBUTIONS

To further investigate the differences between Artificial Neural Networks (ANNs) and Spiking Neural Networks (SNNs), we analyze the number of outliers in the weight distributions across various layers and components. Outliers are defined as weights that significantly deviate from the mean, potentially indicating instability or over-parameterization in specific components.

#### A.10.1 TOTAL NUMBER OF OUTLIERS

As summarized in Table 4, ANNs exhibit significantly more outliers, with over three times as many compared to SNNs across all evaluated layers.

Table 4: Total number of weight outliers across all layers and components.

| Model | Total Outliers |
|-------|----------------|
| ANN   | 18,663         |
| SNN   | 5,834          |

#### A.10.2 LAYER-WISE COMPARISON

To gain deeper insights, we present a detailed layer-wise and component-wise comparison in Table 5, showing the number of outliers for both models. **ANNs exhibit a significantly higher number of outliers**, especially in deeper fully connected layers (fc1, fc2), which may be due to larger weight magnitudes and higher variance. While **SNNs show fewer outliers overall**, reflecting their more compact and tightly regulated weight distributions. Interestingly, in some components (e.g., q_proj, k_proj at Layer 7), SNNs have more outliers than ANNs. This suggests local spikes in weight variability, possibly due to the intrinsic dynamics of spiking updates. The standard deviation of weights (not shown here) is consistently lower in SNNs, reinforcing the observation that they operate within a narrower, more stable range. These findings highlight a fundamental difference in the behavior of ANNs and SNNs: while ANNs may rely on larger weight magnitudes and are more prone to extreme values, SNNs exhibit smoother, biologically plausible weight distributions that reduce the risk of instability.

Table 5: Comparison of weight statistics between ANN and SNN across various layers and components.

| Layer | Component | Model | Mean | Std | Max | Min | Num_Outliers |
|---|---|---|---|---|---|---|---|
| 0 | q_proj | ANN | -3.15E-05 | 7.87E-02 | 0.77 | -0.80 | 596 |
| 0 | q_proj | SNN | 1.30E-04 | 1.14E-02 | 0.06 | -0.06 | **6** |
| 0 | k_proj | ANN | 6.12E-05 | 7.26E-02 | 0.77 | -0.70 | 1154 |
| 0 | k_proj | SNN | -3.36E-05 | 1.10E-02 | 0.06 | -0.06 | **24** |
| 0 | v_proj | ANN | -1.55E-05 | 1.28E-02 | 0.17 | -0.19 | 163 |
| 0 | v_proj | SNN | 2.97E-05 | 1.31E-02 | 0.08 | -0.08 | **34** |
| 0 | out_proj | ANN | -8.64E-06 | 1.30E-02 | 0.78 | -0.62 | 702 |
| 0 | out_proj | SNN | -2.16E-04 | 1.58E-02 | 0.12 | -0.11 | **73** |
| 0 | fc1 | ANN | -3.17E-03 | 2.93E-02 | 0.56 | -1.00 | 5657 |
| 0 | fc1 | SNN | 4.58E-04 | 1.16E-02 | 0.06 | -0.06 | **2** |
| 0 | fc2 | ANN | -1.23E-05 | 2.62E-02 | 1.00 | -1.00 | 716 |
| 0 | fc2 | SNN | 2.74E-04 | 1.24E-02 | 0.06 | -0.11 | **189** |
| 3 | q_proj | ANN | -2.93E-04 | 6.02E-02 | 0.56 | -0.48 | 105 |
| 3 | q_proj | SNN | 1.98E-04 | 1.15E-02 | 0.07 | -0.10 | **57** |
| 3 | k_proj | ANN | 1.19E-04 | 6.16E-02 | 0.45 | -0.46 | 116 |
| 3 | k_proj | SNN | 1.34E-04 | 1.10E-02 | 0.10 | -0.10 | **199** |
| 3 | v_proj | ANN | 1.71E-05 | 2.06E-02 | 0.23 | -0.27 | 60 |
| 3 | v_proj | SNN | -2.36E-04 | 1.39E-02 | 0.09 | -0.09 | **63** |
| 3 | out_proj | ANN | 2.38E-06 | 1.74E-02 | 0.50 | -0.36 | 183 |
| 3 | out_proj | SNN | -2.75E-04 | 1.48E-02 | 0.10 | -0.10 | **79** |
| 3 | fc1 | ANN | -1.81E-03 | 2.49E-02 | 0.86 | -0.73 | 2045 |
| 3 | fc1 | SNN | 2.39E-05 | 1.30E-02 | 0.07 | -0.08 | **38** |
| 3 | fc2 | ANN | -2.44E-06 | 2.67E-02 | 0.64 | -1.06 | 2053 |
| 3 | fc2 | SNN | 4.87E-05 | 1.30E-02 | 0.08 | -0.10 | **137** |
| 7 | q_proj | ANN | -1.80E-04 | 5.98E-02 | 0.53 | -0.54 | 147 |
| 7 | q_proj | SNN | -4.13E-04 | 1.10E-02 | 0.13 | -0.13 | **1484** |
| 7 | k_proj | ANN | -2.34E-05 | 6.03E-02 | 0.74 | -0.68 | 185 |
| 7 | k_proj | SNN | 5.25E-04 | 1.35E-02 | 0.17 | -0.19 | **1004** |
| 7 | v_proj | ANN | 1.22E-05 | 2.02E-02 | 0.13 | -0.14 | 20 |
| 7 | v_proj | SNN | -2.11E-04 | 1.35E-02 | 0.12 | -0.12 | **207** |
| 7 | out_proj | ANN | 1.25E-05 | 1.76E-02 | 0.19 | -0.19 | 93 |
| 7 | out_proj | SNN | -1.91E-04 | 2.03E-02 | 0.19 | -0.17 | **609** |
| 7 | fc1 | ANN | -4.40E-03 | 2.73E-02 | 0.59 | -0.59 | 315 |
| 7 | fc1 | SNN | 6.75E-05 | 1.30E-02 | 0.15 | -0.13 | **224** |
| 7 | fc2 | ANN | -3.44E-05 | 3.00E-02 | 0.38 | -1.00 | 1315 |
| 7 | fc2 | SNN | -8.44E-05 | 1.24E-02 | 0.10 | -0.12 | **204** |
| 11 | q_proj | ANN | -2.52E-04 | 5.57E-02 | 0.50 | -0.49 | 153 |
| 11 | q_proj | SNN | -1.89E-04 | 1.19E-02 | 0.15 | -0.12 | **181** |
| 11 | k_proj | ANN | 5.30E-05 | 5.44E-02 | 0.53 | -0.53 | 366 |
| 11 | k_proj | SNN | 2.98E-04 | 1.17E-02 | 0.19 | -0.11 | **225** |
| 11 | v_proj | ANN | 4.60E-05 | 2.95E-02 | 0.20 | -0.50 | 68 |
| 11 | v_proj | SNN | -5.92E-05 | 1.35E-02 | 0.24 | -0.19 | **268** |
| 11 | out_proj | ANN | -2.00E-05 | 3.15E-02 | 0.89 | -0.88 | 478 |
| 11 | out_proj | SNN | -1.07E-05 | 1.96E-02 | 0.19 | -0.16 | **325** |
| 11 | fc1 | ANN | 4.87E-03 | 2.82E-02 | 0.92 | -1.04 | 473 |
| 11 | fc1 | SNN | -4.00E-05 | 1.37E-02 | 0.07 | -0.09 | **52** |
| 11 | fc2 | ANN | 3.81E-05 | 3.23E-02 | 1.56 | -1.23 | 1500 |
| 11 | fc2 | SNN | 2.15E-05 | 1.50E-02 | 0.12 | -0.11 | **150** |

### A.11 EXTENDING CSSA TO LLAMA ARCHITECTURE

To demonstrate the versatility of our proposed CSSA (Causal Spiking Self-Attention) mechanism, we extend its application beyond the original OPT-based SpikingLLM architecture. Specifically, we implement CSSA on the Llama architecture (Touvron et al., 2023), training two new models: SpikingLLM-v1-Llama with 165M and 1.2B parameters. This expansion validates that CSSA is designed fundamentally for spike-based language modeling, independent of specific architectural choices.

Table 6: Performance comparison of SpikingLLM models across different architectures and timesteps.

| Model | Params (B) | Time Step | Zero-shot Accuracy (%) ↑ | | | | | | | | |
|---|---|---|---|---|---|---|---|---|---|---|---|
| | | | ARC-e | ARC-c | WG | BQ | PIQA | HS | OBQA | HQA | Avg. |
| SpikingLLM-v1-OPT | 0.125 | T=2 | 39.1 | 18.9 | 50.3 | 52.7 | 56.7 | 28.1 | 19.8 | 22.9 | 36.05 |
| SpikingLLM-v1-OPT | 0.125 | T=4 | 39.4 | 19.0 | 51.2 | 53.0 | 57.5 | 29.2 | 19.7 | 23.1 | 36.50 |
| SpikingLLM-v1-Llama | 0.165 | T=2 | 39.3 | 19.2 | 51.3 | 52.8 | 56.8 | 28.3 | 20.3 | 23.2 | 36.40 |
| SpikingLLM-v1-Llama | 0.165 | T=4 | 39.6 | 19.6 | 51.7 | 53.2 | 57.3 | 29.5 | 20.7 | 23.7 | 36.91 |
| SpikingLLM-v1-OPT | 1.300 | T=2 | 45.7 | 23.5 | 54.2 | 56.3 | 62.3 | 40.2 | 24.5 | 24.0 | 41.33 |
| SpikingLLM-v1-OPT | 1.300 | T=4 | 46.3 | 24.3 | 55.6 | 56.8 | 63.4 | 41.7 | 25.2 | 24.3 | 42.19 |
| SpikingLLM-v1-Llama | 1.200 | T=2 | 45.8 | 23.9 | 54.7 | 56.1 | 63.0 | 40.6 | 24.7 | 24.1 | 41.61 |
| SpikingLLM-v1-Llama | 1.200 | T=4 | 46.5 | 24.4 | 55.9 | 56.6 | 63.6 | 41.8 | 25.4 | 24.4 | 42.33 |

The results show consistent performance improvements when increasing timesteps from T=2 to T=4 across all model variants. Notably, the Llama-based models achieve comparable or slightly better results than their OPT-based counterparts, particularly in the 1.2B parameter range where SpikingLLM-Llama (T=4) reaches an average accuracy of 42.33%. This demonstrates that CSSA effectively captures spiking dynamics across different transformer architectures while maintaining competitive language modeling capabilities.

### A.12 COMPARISON WITH QUANTIZED ANNS

To contextualize our contributions, it is crucial to distinguish Spiking Neural Networks (SNNs) from quantized Artificial Neural Networks (ANNs), a common point of comparison. Quantized ANNs achieve efficiency through **spatial discretization**, converting continuous floating-point weights or activations into low-bit, fixed-point formats (e.g., 2-bit, 4-bit). While this approach facilitates model compression and acceleration, the underlying computation remains fundamentally dependent on dense Multiply-Accumulate (MAC) operations. Although 1-bit networks eliminate MACs, they are notoriously difficult to train effectively.

In stark contrast, SNNs leverage **temporal sparse encoding** via binary spikes. A spike (1) or its absence (0) at a given timestep encodes information, enabling event-driven and asynchronous computation. This paradigm allows for substantial energy savings on neuromorphic hardware by exploiting sparsity, a benefit difficult for quantized ANNs to replicate (Horowitz, 2014). Our work harnesses this inherent property of SNNs to significantly reduce inference energy while preserving language generation capabilities.

We provide a direct performance comparison between our 1.3B/1.2B SpikingLLM models and state-of-the-art quantization methods in Table 7. While quantized ANNs like Shao et al. (2023), Kaushal et al. (2024), and Wang et al. (2023) may exhibit a marginal edge in accuracy, we emphasize that these represent fundamentally different methodological paradigms.

Therefore, the primary contribution of our work is not to surpass quantization methods in accuracy, but to pioneer and validate a new, energy-efficient pathway for large language models. Our significance is threefold: (1) We introduce the first train-from-scratch binary-spike-based LLM; (2) We propose the Continuous Spiking Self-Attention (CSSA) mechanism to enable effective causal modeling in SNNs; and (3) We demonstrate the fundamental feasibility of SNN-LLMs, filling a critical

gap in the field and establishing a foundation for future research into scalable, energy-conscious language models.

Table 7: Performance comparison of SpikingLLM with quantized LLMs.

| Model | Params (B) | Zero-shot Accuracy (%) ↑ | | | | | | | | |
|---|---|---|---|---|---|---|---|---|---|---|
| | | ARC-e | ARC-c | WG | BQ | PIQA | HS | OBQA | HQA | Avg. |
| SpikingLLM-v1 (T=2) | 1.3 | 45.7 | 23.5 | 54.2 | 56.3 | 62.3 | 40.2 | 24.5 | 24.0 | 41.33 |
| SpikingLLM-v1 (T=4) | 1.3 | 46.3 | 24.3 | 55.6 | 56.8 | 63.4 | 41.7 | 25.2 | 24.3 | 42.19 |
| SpikingLLM-v1-Llama (T=2) | 1.2 | 45.8 | 23.9 | 54.7 | 56.1 | 63.0 | 40.6 | 24.7 | 24.1 | 41.61 |
| SpikingLLM-v1-Llama (T=4) | 1.2 | 46.5 | 24.4 | 55.9 | 56.6 | 63.6 | 41.8 | 25.4 | 24.4 | 42.33 |
| BitNet (1.58-bit) | 1.3 | 48.7 | 24.1 | 56.8 | 57.4 | 64.2 | 40.6 | 24.6 | 24.9 | 42.66 |
| SmoothQuant (W4A4) | 1.3 | 44.3 | 23.5 | 54.2 | 55.6 | 62.3 | 40.0 | 23.9 | 23.2 | 40.88 |
| OmniQuant (W4A4) | 1.3 | 49.7 | 25.5 | 58.1 | 58.3 | 66.0 | 41.7 | 26.2 | 25.7 | 43.90 |
| TriLM (1.58-bit) | 1.1 | 46.2 | 23.9 | 55.5 | 56.5 | 64.1 | 39.2 | 24.7 | 24.5 | 41.83 |
| TriLM (1.58-bit) | 1.5 | 49.1 | 25.2 | 57.3 | 57.2 | 64.5 | 41.1 | 25.6 | 25.3 | 43.16 |

## A.13 EFFECT OF TRAINING SCALE AND CONVERSATIONAL ABILITY

To validate the scalability of our SpikingLLM, we conducted experiments to assess the impact of increased training data. Our initial 1B and 10B token training runs were conducted under constrained GPU resources, primarily serving as a proof-of-concept. To further probe the potential of our model, we scaled the training for the 125M and 1.3B models to 5B and 25B tokens, respectively. While this scale is still modest compared to standard pre-training regimens, the use of knowledge distillation from a teacher model allows the student SpikingLLM to learn more effectively, mitigating the extensive data requirements typically associated with training from scratch.

Table 8: Performance of SpikingLLM with varying training token counts.

| Model | Params (B) | Tokens (B) | Zero-shot Accuracy (%) ↑ | | | | | | | | |
|---|---|---|---|---|---|---|---|---|---|---|---|
| | | | ARC-e | ARC-c | WG | BQ | PIQA | HS | OBQA | HQA | Avg. |
| SpikingLLM-v1 (T=2) | 0.125 | 1.0 | 39.1 | 18.9 | 50.3 | 52.7 | 56.7 | 28.1 | 19.8 | 22.9 | 36.05 |
| SpikingLLM-v1 (T=2) | 0.125 | 5.0 | 41.4 | 19.2 | 51.4 | 53.4 | 58.2 | 30.2 | 19.9 | 23.1 | 37.10 |
| SpikingLLM-v1 (T=2) | 1.300 | 10.0 | 45.7 | 23.5 | 54.2 | 56.3 | 62.3 | 40.2 | 24.5 | 24.0 | 41.33 |
| SpikingLLM-v1 (T=2) | 1.300 | 25.0 | 48.3 | 26.4 | 57.8 | 58.6 | 65.1 | 44.9 | 27.3 | 26.7 | 44.39 |

As shown in Table 8, increasing the training tokens yields significant performance gains. The average accuracy of the 125M model improved by 1.05% when trained on 5B tokens compared to 1B tokens. More impressively, the 1.3B model's average accuracy increased by 3.06% when scaling from 10B to 25B tokens. These results demonstrate that SpikingLLM is not merely a small-scale proof-of-concept but a model architecture that responds positively and effectively to increased training data, suggesting strong potential for further scaling.

To further investigate the training dynamics, we plot the training loss curves for our models. Figure 14 illustrates the loss progression for the 1.3B model trained on both 10B and 25B tokens. The curves exhibit a smooth and consistent downward trend, indicating that our training process is stable and converges effectively. Notably, the model trained on 25B tokens continues to decrease its loss to a lower final value, corroborating the quantitative performance gains observed in Table 8. This stable convergence behavior across different training scales demonstrates the robustness of our proposed SpikingLLM architecture and the effectiveness of the CSSA mechanism in facilitating the optimization of spike-based language models.

Beyond quantitative metrics, we qualitatively evaluated the conversational abilities of our SpikingLLM-1.3B model trained with 10B and 25B tokens. Table 9 presents sample responses to a set of prompts. The 10B model exhibits basic conversational skills and can maintain a simple

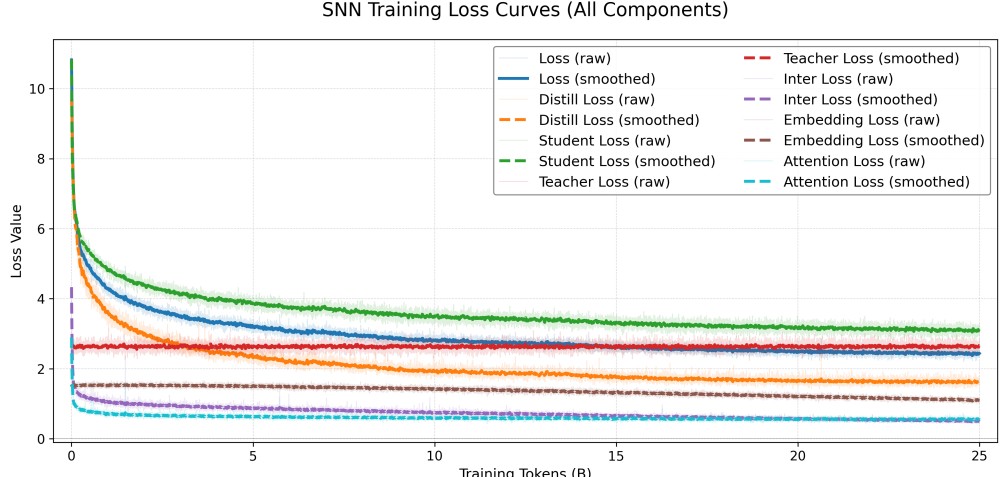

Figure 14: Training loss curves for the SpikingLLM-1.3B model, comparing the 10B and 25B token training regimes. The smooth downward trend confirms stable convergence.

dialogue. However, it fails to answer factual questions correctly, likely due to a lack of exposure to that specific knowledge during its limited training. In contrast, the 25B model provides more accurate and detailed responses, correctly answering the factual question about the U.S. capital and generating more coherent and contextually relevant text. This qualitative improvement aligns with the quantitative gains, underscoring the importance of training scale for developing the factual knowledge and reasoning capabilities of spiking language models.

Table 9: Comparison of conversational abilities for SpikingLLM-v1-1.3B trained on different token counts.

| Prompt | SpikingLLM-v1 (10B Tokens) | SpikingLLM-v1 (25B Tokens) |
| --- | --- | --- |
| *Hi there! How are you doing today?* | Hi! How are you? I'm doing well. How about you? | I'm doing great! I'm just tired as hell. I went to bed at 11pm last night, and now I'm up at 4am. I have a class tomorrow morning at 7:30, so I have to get up and go to class. |
| *Where is the capital of the United States?* | It's in the middle of the country. | Washington, D.C. |
| *How about the cat?* | I think it's a good idea to have a cat in the house. | I have a cat and he doesn't like the camera. I'll send you a picture when he is in the mood. |

### A.14 ANALYSIS OF LONG-CONTEXT DEPENDENCY AND SPARSITY

In the Continuous Spiking Self-Attention (CSSA) mechanism, information propagation relies on spike-triggered updates of the attention state. Consequently, a higher sparsity level—achieved by increasing the firing threshold or reducing the number of timesteps—can theoretically weaken the transmission of long-range signals, potentially impairing the model's ability to model long-distance dependencies.

To empirically evaluate this, we conducted a sliding-window evaluation on the WikiText-2 dataset. We compared our SpikingLLM-1.3B (T=4) and SpikingLLM-Llama-1.2B (T=4) models with varying firing thresholds against the full-precision OPT-1.3B and Llama-3.2-1.2B baselines. The models were tested with context lengths of 512, 1024, 2048, 4096, and 8192 tokens, and the performance

was measured using Perplexity (PPL). For reference, the maximum supported context length for OPT-1.3B is 2048. The results are presented in Table 10.

Table 10: Perplexity (PPL) on WikiText-2 with varying context lengths and firing thresholds. The firing rate (sparsity) is shown in parentheses.

| Model | Params | Threshold | Context Length | | | | |
| --- | --- | --- | --- | --- | --- | --- | --- |
| | | | 512 | 1024 | 2048 | 4096 | 8192 |
| OPT | 1.3B | - | 16.26 | 13.58 | 11.13 | - | - |
| Llama-3.2 | 1.2B | - | 12.93 | 10.96 | 9.76 | 9.02 | 8.54 |
| SpikingLLM-v1 | 1.3B | 0.70 | 36.72 (0.184) | 32.49 (0.183) | 29.34 (0.181) | - | - |
| | | 0.85 | 39.75 (0.177) | 36.30 (0.175) | 32.18 (0.174) | - | - |
| | | 1.00 | 43.17 (0.163) | 39.88 (0.162) | 34.62 (0.160) | - | - |
| SpikingLLM-v1-Llama | 1.2B | 0.70 | 33.53 (0.198) | 29.82 (0.195) | 27.59 (0.194) | 25.32 (0.192) | 23.77 (0.190) |
| | | 0.85 | 37.14 (0.186) | 33.37 (0.185) | 31.26 (0.185) | 28.89 (0.183) | 25.42 (0.181) |
| | | 1.00 | 40.88 (0.174) | 37.25 (0.172) | 34.08 (0.171) | 30.76 (0.171) | 27.19 (0.168) |

The results lead to several key observations. First, as the sparsity increases (i.e., the threshold rises from 0.70 to 1.00), the PPL performance degrades across all context lengths. This confirms our intuition that higher sparsity can impede the flow of information. However, the magnitude of this degradation is moderate, suggesting that increasing sparsity within a certain range does not cause a catastrophic collapse in long-context dependency handling.

And more intriguingly, for a fixed threshold, the PPL consistently decreases as the context length increases. For instance, the SpikingLLM-v1-Llama model with a threshold of 0.70 improves from a PPL of 33.53 at 512 tokens to 23.77 at 8192 tokens. This indicates that the natural increase in sparsity caused by longer sequences has a negligible negative impact on the model's capability. The model retains its ability to extract global information from longer contexts.

Finally, we acknowledge that the current design of CSSA lacks specialized mechanisms for handling extremely long contexts. We are actively working to address this limitation through future research directions, such as adaptive spike scheduling and the integration of long-term memory neurons. We believe these are engineering frontier challenges rather than fundamental obstacles and can be progressively overcome in future work.

