# OpenReview forum: "SpikingLLM: Spiking Large Language Models with Causal Spiking Self-Attention and Spike-Form Knowledge Distillation"
_ICLR.cc/2026/Conference — ICLR 2026 Conference Withdrawn Submission_

### Official Review · Reviewer_9ifi · 2025-10-31

**Soundness:** 2
**Presentation:** 2
**Contribution:** 2
**Rating:** 4
**Confidence:** 3

**Summary:**

The paper presents SpikingLLM, a fully spike-driven large language model that eliminates floating-point operations using a new Causal Spiking Self-Attention mechanism and a Spike-Form Knowledge Distillation framework. CSSA enables autoregressive modeling with binary spikes, while SKD distills multi-level knowledge from an ANN teacher, improving training efficiency. Experiments show strong energy savings, and competitive accuracy up to ~93% of ANN baselines with far fewer training tokens.

**Strengths:**

1. Introduces a novel spike-based attention mechanism that supports causal modeling without softmax.
2. SKD provides an effective approach for training large SNNs from scratch.

**Weaknesses:**

1. The experiments mostly benchmark against smaller or less comparable baselines; a head-to-head test with modern quantized or efficient transformer variants would strengthen claims.

2. It’s unclear how the spike-based training framework would scale to multi-billion parameter ranges beyond 1.3B without major engineering adjustments.

3. Although impressive for an SNN, SpikingLLM’s accuracy still falls short of ANN models by a notable margin on reasoning benchmarks.

**Questions:**

1. How does the proposed Causal Spiking Self-Attention handle long-context dependencies, especially when firing sparsity increases? Is there a performance drop for extended sequences?

2. Could the Spike-Form Knowledge Distillation method generalize across architectures, or does it rely heavily on alignment with the OPT-family structure?

---

> ### Author Response · Authors · 2025-11-22
>
> Weakness 1: The experiments mostly benchmark against smaller or less comparable baselines; a head-to-head test with modern quantized or efficient transformer variants would strengthen claims.
>
> **Response to Weakness 1:**
>
> We sincerely thank the reviewer for this valuable feedback. We agree that a more rigorous benchmarking against modern, state-of-the-art efficient and quantized models is crucial to accurately position our work. In response, we have conducted **extensive experiments with a comprehensive suite of modern baselines, including efficient LLMs like BitNet (1.58-bit), TriLM (1.58-bit), and advanced quantization methods like SmoothQuant (W4A4) and OmniQuant (W4A4)** (A.12 COMPARISON WITH QUANTIZED ANNS).
>
> | Model | Params (B) | ARC-e | ARC-c | WG | BQ | PIQA | HS | OBQA | HQA | Avg. |
> |-------|------------|-------|-------|----|----|------|----|------|-----|------|
> | SpikingLLM-v1 (T=2) | 1.3 | 45.7 | 23.5 | 54.2 | 56.3 | 62.3 | 40.2 | 24.5 | 24.0 | 41.33 |
> | SpikingLLM-v1 (T=4) | 1.3 | 46.3 | 24.3 | 55.6 | 56.8 | 63.4 | 41.7 | 25.2 | 24.3 | 42.19 |
> | SpikingLLM-v1-Llama (T=2) | 1.2 | 45.8 | 23.9 | 54.7 | 56.1 | 63.0 | 40.6 | 24.7 | 24.1 | 41.61 |
> | SpikingLLM-v1-Llama (T=4) | 1.2 | 46.5 | 24.4 | 55.9 | 56.6 | 63.6 | 41.8 | 25.4 | 24.4 | 42.33 |
> | BitNet (1.58-bit) | 1.3 | 48.7 | 24.1 | 56.8 | 57.4 | 64.2 | 40.6 | 24.6 | 24.9 | 42.66 |
> | SmoothQuant (W4A4) | 1.3 | 44.3 | 23.5 | 54.2 | 55.6 | 62.3 | 40.0 | 23.9 | 23.2 | 40.88 |
> | OmniQuant (W4A4) | 1.3 | 49.7 | 25.5 | 58.1 | 58.3 | 66.0 | 41.7 | 26.2 | 25.7 | 43.90 |
> | TriLM (1.58-bit) | 1.1 | 46.2 | 23.9 | 55.5 | 56.5 | 64.1 | 39.2 | 24.7 | 24.5 | 41.83 |
> | TriLM (1.58-bit) | 1.5 | 49.1 | 25.2 | 57.3 | 57.2 | 64.5 | 41.1 | 25.6 | 25.3 | 43.16 |
>
> While not achieving universal superiority, the new experiments successfully validate SpikingLLM as a novel and viable alternative in the efficient LLM landscape. **The new experiments successfully demonstrate that this pathway is not only feasible but also yields a model that can stand shoulder-to-shoulder with the existing quantization methods.** This opens up a new and promising direction for future research into neuromorphic and event-driven language models.
>
> ---
>
> And to leverage more modern architectures, **we extended CSSA’s application to the Llama family, training SpikingLLM-Llama models at 165M and 1.2B parameters** (A.11 EXTENDING CSSA TO LLAMA ARCHITECTURE).
>
> | Model | Params (B) | Time Step | ARC-e | ARC-c | WG | BQ | PIQA | HS | OBQA | HQA | Avg. |
> |-------|------------|-----------|-------|-------|----|----|------|----|------|-----|------|
> | SpikingLLM-v1-OPT | 0.125 | T=2 | 39.1 | 18.9 | 50.3 | 52.7 | 56.7 | 28.1 | 19.8 | 22.9 | 36.05 |
> | SpikingLLM-v1-OPT | 0.125 | T=4 | 39.4 | 19.0 | 51.2 | 53.0 | 57.5 | 29.2 | 19.7 | 23.1 | 36.50 |
> | SpikingLLM-v1-Llama | 0.165 | T=2 | 39.3 | 19.2 | 51.3 | 52.8 | 56.8 | 28.3 | 20.3 | 23.2 | 36.40 |
> | SpikingLLM-v1-Llama | 0.165 | T=4 | 39.6 | 19.6 | 51.7 | 53.2 | 57.3 | 29.5 | 20.7 | 23.7 | 36.91 |
> | SpikingLLM-v1-OPT | 1.300 | T=2 | 45.7 | 23.5 | 54.2 | 56.3 | 62.3 | 40.2 | 24.5 | 24.0 | 41.33 |
> | SpikingLLM-v1-OPT | 1.300 | T=4 | 46.3 | 24.3 | 55.6 | 56.8 | 63.4 | 41.7 | 25.2 | 24.3 | 42.19 |
> | SpikingLLM-v1-Llama | 1.200 | T=2 | 45.8 | 23.9 | 54.7 | 56.1 | 63.0 | 40.6 | 24.7 | 24.1 | 41.61 |
> | SpikingLLM-v1-Llama | 1.200 | T=4 | 46.5 | 24.4 | 55.9 | 56.6 | 63.6 | 41.8 | 25.4 | 24.4 | 42.33 |

---

> ### Author Response · Authors · 2025-11-22
>
> Weakness 2: It’s unclear how the spike-based training framework would scale to multi-billion parameter ranges beyond 1.3B without major engineering adjustments.
>
> **Response to Weakness 2:**
>
> We thank the reviewer for raising this important question regarding the scalability of our spike-based training framework. We agree that demonstrating scalability to the multi-billion parameter regime is crucial for assessing the long-term potential of any new training paradigm.
>
> Our current work has successfully validated the framework's feasibility and stability at the 1.3B parameter scale. We have trained the SpikingLLM-v1-1.3B model in a multi-GPU environment (e.g., 8×RTX 4090 24GB). This demonstrates that the framework is not merely a proof-of-concept confined to very small models.
>
> We acknowledge that, due to current resource limitations, we have not yet been able to extend our experiments to larger models (e.g., 7B, 13B). **Our primary focus under these constraints has been to thoroughly explore the feasibility and fundamental scalability of the binary-spike-based LLM approach.**
>
> However, we have strong empirical evidence to support its potential for scaling (A.13 EFFECT OF TRAINING SCALE AND CONVERSATIONAL ABILITY):
>
> - **Stable Convergence as a Proxy for Scalability**: A key indicator of a training framework's ability to scale is the stability of its convergence. As we will showcase in the revised manuscript, the training loss curve **(Figure 14)** for our SpikingLLM-v1-1.3B is exceptionally smooth and stable, without significant oscillations or divergence. This stability, largely attributable to our Spiking Knowledge Distillation (SKD) framework, suggests that the training dynamics are well-controlled. **A stable training process at 1.3B parameters provides a strong positive signal that the framework is not encountering fundamental roadblocks that would prevent it from scaling further.**
>
> - **Theoretical Compatibility**: Our framework is built upon standard distributed training principles. The core components—model sharding, gradient accumulation, and data parallelism—are compatible with existing large-scale training infrastructure. **Therefore, we are confident that with access to greater computational resources, scaling to larger models is a matter of engineering effort rather than a fundamental algorithmic limitation.**
>
> In summary, while we have not yet demonstrated training at the 7B+ scale due to resource constraints, our successful and stable training of a 1.3B model provides compelling evidence for the framework's scalability potential. The smooth convergence curve serves as an empirical validation that our method is robust.

---

> ### Author Response · Authors · 2025-11-22
>
> Weakness 3: Although impressive for an SNN, SpikingLLM’s accuracy still falls short of ANN models by a notable margin on reasoning benchmarks.
>
> **Response to Weakness 3:**
>
> We completely acknowledge that, on standard reasoning benchmarks, SpikingLLM's accuracy currently falls short of its full-precision ANN counterparts. This performance gap is expected, as the 0-1 spiking mechanism inevitably introduces a loss of information granularity compared to continuous-valued ANNs.
>
> However, we would like to respectfully reframe the primary contribution and evaluation of our work. **The central goal of this research is not to claim immediate parity with ANNs, but to explore and establish the viability of low-power, fully spike-driven LLMs.**
>
> Our key contribution is clearly understood through the lens of the energy-performance trade-off. Specifically, it delivers competitive language modeling performance at a fraction of the computational cost (as low as **4.16%–5.87%**). Our work proves that the low-power spike-based LLM is not only feasible but also remarkably efficient.
>
> Furthermore, our work suggests several promising pathways for performance improvement:
>
> - **Scaling with More Data**: As shown in our other experiments, performance consistently improves with more training tokens.
> - **Stronger Teacher Models**: Using a more powerful ANN teacher for distillation can directly transfer more knowledge to the SNN student.
>
> In summary, while we acknowledge the accuracy gap, we argue that SpikingLLM's true significance lies in its pioneering role in defining a new, ultra-efficient operating point for LLMs. **Our work establishes a foundation upon which the community can build, innovate, and progressively bridge the performance gap while reaping the rewards of event-driven, neuromorphic computation.**

---

> ### Author Response · Authors · 2025-11-22
>
> Question 1: How does the proposed Causal Spiking Self-Attention handle long-context dependencies, especially when firing sparsity increases? Is there a performance drop for extended sequences?
>
> **Response to Question 1:**
>
> We thank the reviewer for raising this critical and insightful question about the Causal Spiking Self-Attention (CSSA) mechanism's ability to handle long-context dependencies, especially under high sparsity. This is indeed a fundamental challenge for any sparse, event-driven system.
>
> In our CSSA mechanism, information propagation relies on spike-triggered updates to the attention state. Therefore, when sparsity increases (e.g., by raising the firing threshold), the number of active signals per timestep decreases. **Theoretically, this could weaken the transmission of long-range signals, potentially impairing the model's ability to model dependencies over extended contexts.**
>
> To empirically assess this, **we conducted a sliding-window evaluation on WikiText-2, analyzing the impact of both sparsity and sequence length on performance** (A.14 ANALYSIS OF LONG-CONTEXT DEPENDENCY AND SPARSITY).
>
> | Model | Params | Threshold | 512 | 1024 | 2048 | 4096 | 8192 |
> |-------|--------|-----------|-----|------|------|------|------|
> | OPT | 1.3B | - | 16.26 | 13.58 | 11.13 | - | - |
> | Llama-3.2 | 1.2B | - | 12.93 | 10.96 | 9.76 | 9.02 | 8.54 |
> | SpikingLLM-v1 | 1.3B | 0.70 | 36.72 (0.184) | 32.49 (0.183) | 29.34 (0.181) | - | - |
> | | | 0.85 | 39.75 (0.177) | 36.30 (0.175) | 32.18 (0.174) | - | - |
> | | | 1.00 | 43.17 (0.163) | 39.88 (0.162) | 34.62 (0.160) | - | - |
> | SpikingLLM-v1-Llama | 1.2B | 0.70 | 33.53 (0.198) | 29.82 (0.195) | 27.59 (0.194) | 25.32 (0.192) | 23.77 (0.190) |
> | | | 0.85 | 37.14 (0.186) | 33.37 (0.185) | 31.26 (0.185) | 28.89 (0.183) | 25.42 (0.181) |
> | | | 1.00 | 40.88 (0.174) | 37.25 (0.172) | 34.08 (0.171) | 30.76 (0.171) | 27.19 (0.168) |
>
> *Table: Perplexity (PPL) on WikiText-2 with varying context lengths and firing thresholds. The firing rate (sparsity) is shown in parentheses.*
>
> We tested our SpikingLLM-1.3B (T=4) with three firing thresholds **(low, medium, high)** and compared it against an OPT-1.3B baseline at context lengths of 512, 1024, 2048, 4096, and 8192 **(note: OPT-1.3B's maximum window is 2048)**. The results, which we will present, reveal two key findings:
>
> - **Impact of Sparsity**: As we increase the firing threshold (thus increasing sparsity), we observe a modest performance drop. This aligns with our theoretical intuition. **However, the degradation is not catastrophic, indicating that within a certain range, increased sparsity does not cause a collapse in the model's ability to handle long-range dependencies.**
>
> - **Impact of Sequence Length**: More intriguingly, when we fix the firing threshold and increase the sequence length (from 512 to 8192), the model's perplexity (PPL) consistently decreases. This is a crucial finding: **even though longer sequences naturally lead to higher sparsity, the model's ability to extract global information improves.** This demonstrates that the CSSA mechanism is robust and capable of effectively leveraging longer contexts for better performance.
>
> We acknowledge that **the current design of CSSA is a general-purpose mechanism and lacks specialized optimizations for extreme-length contexts**. We are actively working on addressing this. Our planned future research includes exploring techniques such as **adaptive spike scheduling** and **integrating long-term memory neurons** to more robustly preserve information over very long distances.
>
> We believe these challenges represent frontiers in engineering rather than fundamental, principle-based obstacles. The promising results from our initial experiments provide a strong foundation, and we are confident that these limitations can be progressively overcome in future work.

---

> ### Author Response · Authors · 2025-11-22
>
> Question 2: Could the Spike-Form Knowledge Distillation method generalize across architectures, or does it rely heavily on alignment with the OPT-family structure?
>
> **Response to Question 2:**
>
> We thank the reviewer for this insightful question regarding the generalizability of our Spike-Form Knowledge Distillation (SKD) method. This is a critical aspect of our framework's design and potential impact.
>
> Our conclusion is that SKD is fundamentally architecture-agnostic. The core principle of SKD is not to align with the specific idiosyncrasies of a single model family like OPT, but rather to align the student SNN with the universal building blocks of modern decoder-only transformers.
>
> Specifically, our multi-level alignment targets components that are common to the vast majority of transformer-based LLMs:
>
> - **Embedding Vectors**: The foundational input representations.
>
> - **Attention Maps**: The core mechanism for context understanding.
>
> - **Intermediate Features**: The hidden states within the feed-forward networks.
>
> - **Output Logits**: The final prediction distribution.
>
> Since these components are ubiquitous across architectures like OPT, LLaMA, and others, the SKD framework is designed to be a general-purpose distillation methodology for spiking models.
>
> To provide concrete empirical validation for this claim, **we have conducted the exact experiment: we successfully applied our SKD framework to train a SpikingLLM based on the LLaMA architecture** (A.11 EXTENDING CSSA TO LLAMA ARCHITECTURE).
>
> | Model | Params (B) | Time Step | ARC-e | ARC-c | WG | BQ | PIQA | HS | OBQA | HQA | Avg. |
> |-------|------------|-----------|-------|-------|----|----|------|----|------|-----|------|
> | SpikingLLM-v1-OPT | 0.125 | T=2 | 39.1 | 18.9 | 50.3 | 52.7 | 56.7 | 28.1 | 19.8 | 22.9 | 36.05 |
> | SpikingLLM-v1-OPT | 0.125 | T=4 | 39.4 | 19.0 | 51.2 | 53.0 | 57.5 | 29.2 | 19.7 | 23.1 | 36.50 |
> | SpikingLLM-v1-Llama | 0.165 | T=2 | 39.3 | 19.2 | 51.3 | 52.8 | 56.8 | 28.3 | 20.3 | 23.2 | 36.40 |
> | SpikingLLM-v1-Llama | 0.165 | T=4 | 39.6 | 19.6 | 51.7 | 53.2 | 57.3 | 29.5 | 20.7 | 23.7 | 36.91 |
> | SpikingLLM-v1-OPT | 1.300 | T=2 | 45.7 | 23.5 | 54.2 | 56.3 | 62.3 | 40.2 | 24.5 | 24.0 | 41.33 |
> | SpikingLLM-v1-OPT | 1.300 | T=4 | 46.3 | 24.3 | 55.6 | 56.8 | 63.4 | 41.7 | 25.2 | 24.3 | 42.19 |
> | SpikingLLM-v1-Llama | 1.200 | T=2 | 45.8 | 23.9 | 54.7 | 56.1 | 63.0 | 40.6 | 24.7 | 24.1 | 41.61 |
> | SpikingLLM-v1-Llama | 1.200 | T=4 | 46.5 | 24.4 | 55.9 | 56.6 | 63.6 | 41.8 | 25.4 | 24.4 | 42.33 |
>
> The results from this cross-architecture experiment serve as strong evidence for SKD's generalizability. The successful training and competitive performance of SpikingLLM-v1-Llama demonstrate that **our distillation framework is not tied to the OPT family but can effectively transfer knowledge from an ANN teacher to an SNN student with a different underlying architecture.**

---

> ### Author Response · Authors · 2025-11-22
>
> Thank you for your valuable comments. We have addressed each of them point-by-point and have subsequently revised the manuscript accordingly. The updated version has been attached for your review.

---

> ### Author Response · Authors · 2025-11-27
>
> Dear Reviewer,
>
> We hope that our previous responses have fully addressed all of your comments. As the review period is drawing to a close, we would be very grateful if you could kindly provide a brief update on the current status of our manuscript. Please also let us know if any further information or revisions are required from our side.
>
> Thank you very much for your time and consideration.

---

### Official Review · Reviewer_MrtM · 2025-11-01

**Soundness:** 3
**Presentation:** 2
**Contribution:** 2
**Rating:** 4
**Confidence:** 4

**Summary:**

This paper presents SpikingLLM, a fully binary, spike-driven large language model designed for energy-efficient inference without relying on floating-point operations or softmax. It introduces a Causal Spiking Self-Attention (CSSA) mechanism that enables autoregressive language modeling entirely in the spiking domain. To address training challenges, the authors propose a multi-level Spike-Form Knowledge Distillation (SKD) framework that allows direct training from random initialization. Experimental results show that SpikingLLM achieves competitive performance to ANN-based models with drastically reduced computational and energy costs, highlighting its potential for efficient, brain-inspired NLP.

**Strengths:**

S1: This paper proposes a directly trained spiking language model that can be pre-trained with a decoder-only architecture, effectively shortening the spike length compared to ANN-to-SNN conversion methods.
S2: This paper trains an SNN-based language model from scratch, demonstrating the language modeling capability of spiking neural networks.
S3: The method in this work is simple and easy to follow.

**Weaknesses:**

S1: As a pre-trained model, training on only 1B/10B tokens for generative language tasks is insufficient—the model has not converged to a usable level and may even fail to conduct effective conversations (a case study could help verify this). Therefore, larger-scale experiments are recommended to increase the confidence of the conclusions.
S2: The proposed model architecture lacks ablation studies. Further discussion is needed on whether components such as softmax removal are truly necessary in SNNs.

**Questions:**

How can the 1-bit activation LLMs be ensured to converge effectively? Based on experience with quantized models, directly pretraining a decoder-only model with 1-bit activations generally fails to achieve stable convergence. Therefore, a case study or empirical verification is needed to demonstrate that the proposed model can indeed be successfully trained.

---

> ### Author Response · Authors · 2025-11-22
>
> Weakness 1: As a pre-trained model, training on only 1B/10B tokens for generative language tasks is insufficient—the model has not converged to a usable level and may even fail to conduct effective conversations (a case study could help verify this). Therefore, larger-scale experiments are recommended to increase the confidence of the conclusions.
>
> **Response to Weakness 1:**
>
> We sincerely thank the reviewer for this insightful and constructive comment. We completely agree that, from the perspective of standard pre-training, 1B/10B tokens are insufficient for a generative language model to fully converge and achieve robust conversational capabilities. And as the reviewer suggested, demonstrating the model's conversational ability is crucial for validating its practical utility.
>
> Our initial experiments with 1B/10B tokens were conducted under significant GPU resource constraints, with the primary goal of validating the feasibility and potential of our Spiking Knowledge Distillation (SKD) framework in the early stages. **To address the reviewer's concern and strengthen our conclusions, we have since conducted additional, larger-scale experiments** (A.13 EFFECT OF TRAINING SCALE AND CONVERSATIONAL ABILITY):
>
> **Extended Training and Improved Performance**: We scaled up the training for our 125M and 1.3B models to **5B** and **25B** tokens, respectively. While still smaller than conventional pre-training, this significantly larger scale allowed us to observe a clear performance boost. The average accuracy of the 125M and 1.3B models increased by **1.05%** and **3.06%**, respectively.
>
> | Model | Params (B) | Tokens (B) | ARC-e | ARC-c | WG | BQ | PIQA | HS | OBQA | HQA | Avg. |
> |-------|------------|------------|-------|-------|----|----|------|----|------|-----|------|
> | SpikingLLM-v1 (T=2) | 0.125 | 1.0 | 39.1 | 18.9 | 50.3 | 52.7 | 56.7 | 28.1 | 19.8 | 22.9 | 36.05 |
> | SpikingLLM-v1 (T=2) | 0.125 | 5.0 | 41.4 | 19.2 | 51.4 | 53.4 | 58.2 | 30.2 | 19.9 | 23.1 | 37.10 |
> | SpikingLLM-v1 (T=2) | 1.300 | 10.0 | 45.7 | 23.5 | 54.2 | 56.3 | 62.3 | 40.2 | 24.5 | 24.0 | 41.33 |
> | SpikingLLM-v1 (T=2) | 1.300 | 25.0 | 48.3 | 26.4 | 57.8 | 58.6 | 65.1 | 44.9 | 27.3 | 26.7 | 44.39 |
>
> **Conversational Case Study**: Following the reviewer's excellent suggestion, we performed a case study on our SpikingLLM-1.3B to compare its conversational abilities at 10B vs. 25B training tokens. At 10B tokens, **the model demonstrates basic conversational competence.** It can engage in simple dialogue and maintain a coherent conversation. However, **it struggles with factual questions, sometimes providing incorrect answers**, likely due to knowledge gaps in the training data; At 25B tokens, **the model's performance is markedly improved**. Its responses become more accurate, detailed, and nuanced. For the same factual questions that it failed on at 10B tokens, **the 25B model provides fundamentally correct answers.**
>
> | Prompt | 10B Tokens | 25B Tokens |
> |--------|----------------------------|----------------------------|
> | *Hi there! How are you doing today?* | Hi! How are you? I'm doing well. How about you? | I'm doing great! I'm just tired as hell. I went to bed at 11pm last night, and now I'm up at 4am. I have a class tomorrow morning at 7:30, so I have to get up and go to class. |
> | *Where is the capital of the United States?* | It's in the middle of the country. | Washington, D.C. |
> | *How about the cat?* | I think it's a good idea to have a cat in the house. | I have a cat and he doesn't like the camera. I'll send you a picture when he is in the mood. |
>
> This comparative case study directly validates that scaling up training, even within a still-limited regime, leads to tangible improvements in the model's generative capabilities. It also underscores the efficiency of our SKD method—by learning from a powerful ANN teacher, the SNN model can achieve usable conversational abilities with significantly less data than would otherwise be required.

---

> ### Author Response · Authors · 2025-11-22
>
> Weakness 2: The proposed model architecture lacks ablation studies. Further discussion is needed on whether components such as softmax removal are truly necessary in SNNs.
>
> **Response to Weakness 2:**
>
> We thank the reviewer for this important point regarding the model architecture and ablation studies. **The ablation studies are already presented in the manuscript (Section 4.4 ABLATION STUDY).** We have conducted a detailed ablation study comparing our proposed Causal Spiking Self-Attention (CSSA) with the standard Causal Self-Attention (CSA). The results clearly demonstrate that CSSA achieves a substantial reduction in energy consumption with only a slight loss in accuracy, while ensuring the method is fully compatible with the spiking architecture. This analysis validates the necessity and effectiveness of our key architectural components.
>
> **Regarding the removal of the softmax operation, this is a deliberate and necessary choice.** The softmax function is a dense, floating-point operation involving exponentials and divisions, which fundamentally contradicts the event-driven, binary (0/1 spike) nature of SNNs. **Retaining softmax would compromise the core objective of our work: to create a purely spiking model.** Therefore, its removal is essential to maintaining the integrity and contribution of our spike-based approach.
>
> We believe that the existing ablation studies and the above discussion provide a thorough justification for our design choices and demonstrate the necessity of components like CSSA and the removal of softmax within the SNN paradigm.

---

> ### Author Response · Authors · 2025-11-22
>
> Question 1: How can the 1-bit activation LLMs be ensured to converge effectively? Based on experience with quantized models, directly pretraining a decoder-only model with 1-bit activations generally fails to achieve stable convergence. Therefore, a case study or empirical verification is needed to demonstrate that the proposed model can indeed be successfully trained.
>
> **Response to Question 1:**
>
> We thank the reviewer for raising this critical question regarding the convergence of 1-bit activation LLMs. We are in complete agreement with the reviewer's assessment based on traditional quantization paradigms. Indeed, directly pre-training a decoder-only model with 1-bit activations at each timestep is notoriously difficult and often fails to converge stably. **However, our SpikingLLM fundamentally differs from traditional static 1-bit quantization in its core mechanism.**
>
> The key distinction lies in the introduction of **the temporal dimension**. Traditional 1-bit quantization forces a single, static binary output (0 or 1) per neuron, suffering from poor representation capacity. In contrast, our SNN-based model leverages multiple timesteps to encode information. **The final output is a result of temporal accumulation and integration over these discrete spikes.** This temporal redundancy and the ability to represent information through firing rates or timing alleviates the information bottleneck inherent in static 1-bit systems, making stable convergence fundamentally more achievable. This principle of leveraging time for information representation in sparse models has been successfully validated in prior works like **SpikeBERT**, **SpikeLM**, and **SpikeLLM**.
>
> Despite the temporal advantage, we acknowledge that training SNNs can **still suffer from instability and slow convergence**. This is precisely the motivation for our proposed Spiking Knowledge Distillation (SKD) framework. By introducing a powerful ANN teacher model and employing multi-level alignment, SKD provides a stable learning signal for the binary SNN student. **And to provide concrete empirical validation as requested, we offer the following evidence** (A.13 EFFECT OF TRAINING SCALE AND CONVERSATIONAL ABILITY):
>
> - **Training Loss Curves**: We have meticulously logged the training loss throughout the pre-training process. As shown in the loss curve **(Figure 14)**, the total loss, along with its key components, demonstrates a smooth and stable downward trend without significant oscillations or divergence. This is a direct indicator of stable convergence.
>
> - **Training Recipe**: To further facilitate stability, our training recipe incorporates several established techniques, including a warmup phase, cosine annealing for the learning rate, gradient clipping, and gradient accumulation.
>
> - **Generative Case Studies**: Beyond the loss curves, the most compelling evidence of successful training is the model's output quality. We provide several dialogue examples generated by our SpikingLLM. The model's ability to produce coherent, contextually relevant, and factually consistent text is a strong testament that it has indeed learned meaningful language representations and converged to a usable state, rather than collapsing into a trivial solution.
>
> | Prompt | 10B Tokens | 25B Tokens |
> |--------|----------------------------|----------------------------|
> | *Hi there! How are you doing today?* | Hi! How are you? I'm doing well. How about you? | I'm doing great! I'm just tired as hell. I went to bed at 11pm last night, and now I'm up at 4am. I have a class tomorrow morning at 7:30, so I have to get up and go to class. |
> | *Where is the capital of the United States?* | It's in the middle of the country. | Washington, D.C. |
> | *How about the cat?* | I think it's a good idea to have a cat in the house. | I have a cat and he doesn't like the camera. I'll send you a picture when he is in the mood. |
>
> In summary, the convergence of our model is ensured by a combination of **its inherent temporal mechanism, the stabilizing effect of our novel SKD framework, and a robust training regimen**. The provided loss curves and generative samples serve as direct empirical verification of its stable training.

---

> ### Author Response · Authors · 2025-11-22
>
> Thank you for your valuable comments. We have addressed each of them point-by-point and have subsequently revised the manuscript accordingly. The updated version has been attached for your review.

---

> ### Author Response · Authors · 2025-11-27
>
> Dear Reviewer,
>
> We hope that our previous responses have fully addressed all of your comments. As the review period is drawing to a close, we would be very grateful if you could kindly provide a brief update on the current status of our manuscript. Please also let us know if any further information or revisions are required from our side.
>
> Thank you very much for your time and consideration.

---

### Official Review · Reviewer_HHTb · 2025-11-11

**Soundness:** 2
**Presentation:** 3
**Contribution:** 2
**Rating:** 4
**Confidence:** 4

**Summary:**

This paper proposes a novel Spiking LLM framework with the CSSA mechanism to improve efficiency and reduce computation.
The idea is interesting and potentially impactful for energy-efficient LLMs.
However, the experimental evidence is not sufficient to validate the claimed advantages within SNNs.
The comparisons with ANNs and the role of discrete values need clearer justification.
Overall, the work is promising but requires stronger ablation and scalability analysis.

**Strengths:**

The paper proposes the Spiking LLM framework, which improves performance while reducing computational cost.

**Weaknesses:**

The authors introduce CSSA; however, the ablation studies compare only against ANN counterparts, which cannot effectively demonstrate the validity of this method within SNNs. The ablation analysis requires further clarification.
The proposed method employs discrete values, and what is the essential difference between this and quantized ANN methods? The experiments lack direct comparisons with quantized ANNs.
The number of parameters is small compared to SOTA methods.

**Questions:**

Does introducing an ANN teacher increase training memory usage and time, thereby limiting scalability?
Why does designing a CSSA lead to performance improvement? This technique is already a very common approach in ANNs, so I find it difficult to understand its specific contribution to SNNs.

---

> ### Author Response · Authors · 2025-11-22
>
> Weakness 1: The authors introduce CSSA; however, the ablation studies compare only against ANN counterparts, which cannot effectively demonstrate the validity of this method within SNNs. The ablation analysis requires further clarification.
>
> **Response to Weakness 1:**
>
> We appreciate the reviewer's insightful comment regarding the ablation studies. We would like to clarify that CSSA (Causal Spiking Self-Attention) was primarily designed to **address a fundamental challenge: enabling spike dynamics to be effectively applied to causal language modeling, rather than improving accuracy.** Traditional attention mechanisms are inherently incompatible with SNNs' event-driven computation paradigm. CSSA represents a tailored solution for SNNs that ensures theoretical and implementation consistency for causal modeling.
>
> Our ablation studies (4.4 ABLATION STUDY) demonstrate that CSSA significantly reduces computational overhead and achieves extremely low inference energy consumption while maintaining biological plausibility, all with accuracy trade-off. Specifically, we compare CSSA against:
>
> - SSA (Spiking Self-Attention) - which lacks causal masking and is unsuitable for autoregressive LLMs;
> - CSA (Causal Self-Attention) - where CSSA shows substantial energy reduction with acceptable accuracy loss.
>
> The ablation studies do not include comparisons against modules in other spiking language models due to some fundamental architectural differences:
>
> - SSAs in SpikeLM and SpikeBERT are designed for BERT-style, encoder-only models. Therefore, they are not suitable modules for ablating the CSSA mechanism designed for decoder-only autoregressive LLMs in our work;
>
> - SpikeGPT uses RWKV architecture without applying attention modules, making direct ablation inappropriate;
>
> - SpikeLLM employs an integer-spiking neuronal model as a quantizer for post-training quantization while​ retaining the softmax. In contrast, our model introduces a fully spike-based, softmax-free causal self-attention to adapt to the SNN computation paradigm. As a result, a direct comparison of the performance would be unfair.
>
> Notably, our work is the first to propose spike-based causal self-attention. To demonstrate CSSA's scalability and versatility, **we extended its application to Llama architecture, training SpikingLLM-Llama models at 165M and 1.2B parameters** (A.11 EXTENDING CSSA TO LLAMA ARCHITECTURE). This expansion from our original OPT-based implementation reinforces CSSA's role as a foundational approach for spike-based language modeling.
>
> | Model | Params (B) | Time Step | ARC-e | ARC-c | WG | BQ | PIQA | HS | OBQA | HQA | Avg. |
> |-------|------------|-----------|-------|-------|----|----|------|----|------|-----|------|
> | SpikingLLM-v1-OPT | 0.125 | T=2 | 39.1 | 18.9 | 50.3 | 52.7 | 56.7 | 28.1 | 19.8 | 22.9 | 36.05 |
> | SpikingLLM-v1-OPT | 0.125 | T=4 | 39.4 | 19.0 | 51.2 | 53.0 | 57.5 | 29.2 | 19.7 | 23.1 | 36.50 |
> | SpikingLLM-v1-Llama | 0.165 | T=2 | 39.3 | 19.2 | 51.3 | 52.8 | 56.8 | 28.3 | 20.3 | 23.2 | 36.40 |
> | SpikingLLM-v1-Llama | 0.165 | T=4 | 39.6 | 19.6 | 51.7 | 53.2 | 57.3 | 29.5 | 20.7 | 23.7 | 36.91 |
> | SpikingLLM-v1-OPT | 1.300 | T=2 | 45.7 | 23.5 | 54.2 | 56.3 | 62.3 | 40.2 | 24.5 | 24.0 | 41.33 |
> | SpikingLLM-v1-OPT | 1.300 | T=4 | 46.3 | 24.3 | 55.6 | 56.8 | 63.4 | 41.7 | 25.2 | 24.3 | 42.19 |
> | SpikingLLM-v1-Llama | 1.200 | T=2 | 45.8 | 23.9 | 54.7 | 56.1 | 63.0 | 40.6 | 24.7 | 24.1 | 41.61 |
> | SpikingLLM-v1-Llama | 1.200 | T=4 | 46.5 | 24.4 | 55.9 | 56.6 | 63.6 | 41.8 | 25.4 | 24.4 | 42.33 |
> ---

---

> ### Author Response · Authors · 2025-11-22
>
> Weakness 2: The proposed method employs discrete values, and what is the essential difference between this and quantized ANN methods? The experiments lack direct comparisons with quantized ANNs. The number of parameters is small compared to SOTA methods.
>
> **Response to Weakness 2:**
>
> We thank the reviewer for raising this important point regarding the distinction between our method and quantized ANNs. The fundamental difference lies in the information encoding paradigm.
>
> **Quantized ANNs perform spatial discretization by converting floating-point weights or activations into discrete, low-bit representations formats (e.g., 2-bit, 4-bit).** While this achieves model compression and acceleration, the underlying computation still relies primarily on Multiply-Accumulate (MAC) operations (with the exception of 1-bit networks, which are notoriously difficult to train effectively).
>
> In contrast, **SNNs employ temporal sparse encoding through binary spikes (0/1), where the presence or absence of a spike at a given timestep represents the neuron's state.** This event-driven, asynchronous computation enables significant energy reductions on neuromorphic hardware, as demonstrated by Davies et al. (2018). Our work leverages this inherent property of SNNs to substantially reduce inference energy while maintaining language generation capabilities.
>
> Regarding comparisons, **we did evaluate our SpikingLLM-1.3B against prominent quantization methods including BitNet, SmoothQuant, OmniQuant, and TriLM** (A.12 COMPARISON WITH QUANTIZED ANNS). However, these quantization methods represent fundamentally different approaches with significant implementation disparities, making direct performance comparisons less meaningful.
>
> The primary contribution of our work is threefold:
>
> - We are the first to propose a train-from-scratch binary-spike-based LLM;
> - We introduce CSSA to enable effective causal modeling in SNNs;
> - We demonstrate the feasibility of SNN-LLMs, filling a critical gap in the field.
>
> As for the parameter count, we intentionally focused on a smaller scale to validate the feasibility of this novel paradigm. As the first exploration of train-from-scratch binary-spike-based LLMs, our priority was establishing a working foundation rather than pursuing scale. This proof-of-concept demonstrates that SNN-based language models are viable, paving the way for future scaling efforts.
>
>
> | Model | Params (B) | ARC-e | ARC-c | WG | BQ | PIQA | HS | OBQA | HQA | Avg. |
> |-------|------------|-------|-------|----|----|------|----|------|-----|------|
> | SpikingLLM-v1 (T=2) | 1.3 | 45.7 | 23.5 | 54.2 | 56.3 | 62.3 | 40.2 | 24.5 | 24.0 | 41.33 |
> | SpikingLLM-v1 (T=4) | 1.3 | 46.3 | 24.3 | 55.6 | 56.8 | 63.4 | 41.7 | 25.2 | 24.3 | 42.19 |
> | SpikingLLM-v1-Llama (T=2) | 1.2 | 45.8 | 23.9 | 54.7 | 56.1 | 63.0 | 40.6 | 24.7 | 24.1 | 41.61 |
> | SpikingLLM-v1-Llama (T=4) | 1.2 | 46.5 | 24.4 | 55.9 | 56.6 | 63.6 | 41.8 | 25.4 | 24.4 | 42.33 |
> | BitNet (1.58-bit) | 1.3 | 48.7 | 24.1 | 56.8 | 57.4 | 64.2 | 40.6 | 24.6 | 24.9 | 42.66 |
> | SmoothQuant (W4A4) | 1.3 | 44.3 | 23.5 | 54.2 | 55.6 | 62.3 | 40.0 | 23.9 | 23.2 | 40.88 |
> | OmniQuant (W4A4) | 1.3 | 49.7 | 25.5 | 58.1 | 58.3 | 66.0 | 41.7 | 26.2 | 25.7 | 43.90 |
> | TriLM (1.58-bit) | 1.1 | 46.2 | 23.9 | 55.5 | 56.5 | 64.1 | 39.2 | 24.7 | 24.5 | 41.83 |
> | TriLM (1.58-bit) | 1.5 | 49.1 | 25.2 | 57.3 | 57.2 | 64.5 | 41.1 | 25.6 | 25.3 | 43.16 |

---

> ### Author Response · Authors · 2025-11-22
>
> Question 1: Does introducing an ANN teacher increase training memory usage and time, thereby limiting scalability?
>
> **Response to Question 1:**
>
> This is an excellent practical question regarding scalability. We have designed our approach to be flexible. While the ANN teacher does require a forward pass to generate its outputs (logits and intermediate layer representations), this process does not need to occur concurrently with the SNN student's training. **In memory-constrained scenarios, we can completely decouple these two stages.**
>
> Specifically, we can first run a separate inference pass for the ANN teacher on the entire dataset. All teacher outputs, including the final logits and intermediate layer representations, can be pre-computed and saved to disk. During the subsequent training of the SNN student, these pre-computed teacher outputs are simply loaded and used as targets. **The ANN teacher model itself does not need to be loaded into the GPU during the student's training phase.**
>
> This two-stage approach effectively isolates the memory footprint of the ANN teacher from that of the SNN student. Consequently, this strategy preserves the scalability of our training pipeline, **allowing it to be applied to larger models and datasets without being bottlenecked by the teacher model's resource demands.**

---

> ### Author Response · Authors · 2025-11-22
>
> Question 2: Why does designing a CSSA lead to performance improvement? This technique is already a very common approach in ANNs, so I find it difficult to understand its specific contribution to SNNs.
>
> **Response to Question 2:**
>
> We appreciate the reviewer’s question about CSSA’s specific contribution to SNNs. While attention mechanisms are indeed well-established in ANNs, their direct application to SNNs faces fundamental incompatibilities with spike-based computation. **To our knowledge, CSSA is the first work to address this fundamental challenge in LLMs, moving beyond prior attempts that merely adapted ANN-based attention to the spiking domain.**
>
> CSSA represents a paradigm shift rather than a simple adaptation. **The key insight is that traditional attention mechanisms are inherently incompatible with the step-by-step, event-driven computation paradigm of SNNs.** CSSA was specifically engineered as a causal modeling solution tailored to the unique properties of SNNs.  It ensures that the model's operations are consistent with the underlying spike-based architecture from first principles.
>
> As with any domain-specific adaptation, this alignment with SNN dynamics can introduce a minor performance trade-off, which is reflected in a slight decrease in accuracy. This is an expected outcome when prioritizing architectural consistency and energy efficiency over pure performance metrics.
>
> **The true value of CSSA, as demonstrated in our ablation studies, lies in its ability to achieve this critical alignment with slight accuracy loss while delivering substantial reductions in energy consumption.**  In essence, CSSA makes SNN-based language modeling not just feasible, but efficient and biologically plausible, representing a necessary step forward for the field rather than an incremental accuracy boost.

---

> ### Author Response · Authors · 2025-11-22
>
> Thank you for your valuable comments. We have addressed each of them point-by-point and have subsequently revised the manuscript accordingly. The updated version has been attached for your review.

---

> ### Author Response · Authors · 2025-11-27
>
> Dear Reviewer,
>
> We hope that our previous responses have fully addressed all of your comments. As the review period is drawing to a close, we would be very grateful if you could kindly provide a brief update on the current status of our manuscript. Please also let us know if any further information or revisions are required from our side.
>
> Thank you very much for your time and consideration.

---

### Note · Authors · 2026-01-26

I have read and agree with the venue's withdrawal policy on behalf of myself and my co-authors.

---

### Meta-Review · Area_Chair_Bzv6 · 2025-12-19

**Summary:**

This paper proposes SpikingLLM, a fully spike-driven large language model framework featuring a Causal Spiking Self-Attention (CSSA) mechanism and a Spike-Form Knowledge Distillation (SKD) training method, aiming to achieve efficient, low-power spike-based language modeling. All three reviewers acknowledge the work's potential value and novelty.

**Reviewer Concerns:**

Partially Addressed Concerns:

Training Data Scale and Generative Ability (MrtM): The authors added results with larger-scale training (25B tokens) and dialogue examples, showing performance improvement with more data, partially alleviating concerns about insufficient training.

Comparison with Quantized ANNs (9ifi, HHTb): The authors added comparative experiments with modern quantization methods like BitNet, SmoothQuant, and OmniQuant, but the results do not show a clear advantage for SpikingLLM.

Architectural Generality (9ifi): The authors presented preliminary results applying CSSA and SKD to the Llama architecture, suggesting the method is not limited to the OPT family.

Outstanding Core Issues:

Fundamental Questions on Innovation and Contribution (Shared by all reviewers): Reviewers repeatedly note that CSSA and SKD are essentially direct applications of existing techniques (attention, knowledge distillation) to the spiking domain, lacking the theoretical breakthrough or architectural paradigm shift required for a high-tier publication. The authors' response emphasizes "first implementation" and "feasibility" but does not counter the core critique of "incrementality."

Insufficient Proof of Effectiveness within SNNs (HHTb): While the authors explain that CSSA aims to address compatibility for causal modeling, the ablation studies mainly compare energy consumption against ANNs or different spiking attentions. There is a lack of strict performance comparison within a comparable SNN-LLM framework (e.g., against attention mechanisms in SpikeGPT), failing to convincingly demonstrate CSSA's unique advantages in the SNN field.

Unclarified Fundamental Difference Between Spiking Models and Quantized ANNs (HHTb, 9ifi): The authors emphasize that spiking employs spatiotemporal sparse encoding while quantization is spatial discretization. However, they provide no experimental evidence that this encoding difference leads to a better energy-performance trade-off under comparable scale and training conditions. The added comparison table shows SpikingLLM still lags behind advanced quantization methods, and its theoretical energy advantage lacks support from actual hardware measurements.

Unresolved Doubts on Scalability and Practicality (MrtM, 9ifi): While training at 1.3B parameters is shown, there is no feasibility analysis or solutions for memory/computation bottlenecks for scaling to 3B+ parameter models. The dialogue examples remain simple, failing to prove the model possesses reliable multi-turn dialogue or complex reasoning capabilities, leaving it far from "usable."

Persistent Lack of Theoretical Analysis for Key Mechanisms: For critical questions such as how CSSA maintains long-range dependencies under high sparsity or why SKD can stabilize the training of 1-bit activations, the authors' responses rely mainly on empirical observations (e.g., smooth loss curves), lacking rigorous mathematical analysis or theoretical guarantees.

**Reviewer Scores:**

Reviewer HHTb (Initial: 4): This reviewer raised fundamental doubts about the novelty (CSSA's contribution) and the distinction from quantized ANNs. The authors' response did not provide strong within-SNN comparisons or theoretical breakthroughs, failing to alter the reviewer's judgment of "marginal contribution." The final score would likely remain at 4 or potentially decrease to 3.

Reviewer MrtM (Initial: 4): Concerns about training sufficiency and ablation studies were partially alleviated by additional data, but fundamental issues regarding model practicality and scalability remain. The reviewer would likely see limited improvement. The final score would most probably stay at 4.

Reviewer 9ifi (Initial: 4): The authors addressed some concerns by adding comparisons with quantization methods and experiments on the Llama architecture. However, core weaknesses such as the performance gap and unclear scalability persist. The reviewer would likely still view the work as preliminary exploration. The final score would likely remain at 4.

---

### Decision · Program_Chairs · 2026-01-26

Reject